# Mineralogical-Petrographic and Physical-Mechanical Features of the Construction Stones in Punic and Roman Temples of Antas (SW Sardinia, Italy): Provenance of the Raw Materials and Conservation State

**Stefano Columbu** , **Emanuela Gaviano, Luca Giacomo Costamagna** and **Dario Fancello** *

Dipartimento di Scienze Chimiche e Geologiche, University of Cagliari, Cittadella Universitaria di Monserrato, Monserrato, 09042 Cagliari, Italy; columbus@unica.it (S.C.); gaviano.emanuela@gmail.com (E.G.); lucakost@unica.it (L.G.C.)

* Correspondence: dario.fancello@unica.it

**Abstract:** The Antas site (SW Sardinia, Italy) is of fundamental cultural importance because it testifies the presence of Nuragic, Punic and Roman civilizations from the second millennium to the third century BC. This work focuses on the Punic and the Roman temples and aims to define their conservation state and provenance of construction materials through their minero-petrographic and physical-mechanical characterization. In addition, artificial geomaterials used in restoration works comprising a partial anastylosis and a consolidation intervention on the monument, were investigated to evaluate the aesthetic, petrographic and petrophysical compatibility with the original materials. The results indicate that Punic builders preferred to use a porous sandstone coming from at least few kilometres away from the site. By contrast, Roman builders opted for the use of the less porous and harder local metadolostones, more difficult to quarry and to hew but promptly available in the surrounding area. The Roman temple still preserves decorative architectural elements (as the Pronao threshold and the mosaic tesserae) whose source is definitely not local, suggesting the import of these materials. As regards artificial materials, a new material was found within the Punic temple consisting of a sandstone-like rock (i.e., lime based sandy-conglomeratic geomaterial) and characterized by higher mechanical strength and lower porosity.

**Keywords:** metadolostone; limestone; sandstone; mortar-stone; sulcis-iglesiente; stone; Punic-Roman; provenance; ancient quarry; physical-mechanical properties

## 1. Introduction and Aims of Research

The Antas valley site assumes a great archaeological-cultural significance in relation to the research of the temple of *Sardus Pater*, the location of which has been for many centuries one of the most important open questions of ancient Sardinia. Since the time of the geographer Ptolemy, dozens of coastal sites and some inhabited areas of the hinterland have been listed as possible locations of the temple. Nevertheless, the location of the *Sardus Pater* temple remained uncertain until the 19th century [1].

Furthermore, the Antas valley was traversed by a road that connected the Sulcis sub-region, an area of strategic and economic importance thanks to the widespread occurrence of metal ore deposits [2], to the area of *Tharros* and to northern Sardinia (e.g., *Turris*, today Porto Torres). This route was widely used by Romans but was presumably opened since the Punic phase (third century BC) given that Roman roads were commonly based on already existing routes (often Punic roads). Thus, it is reasonable to assume that the temple was located along this main road. This hypothesis was supported by the analysis of the geomorphological, geological and archaeological data of the Iglesiente area, where it is known that the mining exploitation of the widespread metal ore deposits (mainly lead and silver) dates back to ancient times [1–4]. Several geological studies have revealed that in

the Fluminimaggiore area, in the surroundings of Antas, as well as in other areas of the Iglesiente, ore deposits containing lead and silver widely occur ([5], and references therein). In this area, silver is easily exploitable given the conspicuous exposed outcrops [2], and extraction is possible even without costly quarrying operations. This evidence allow us to hypothesize that the position of the temple/sanctuary could be linked to the exploitation of silver, a precious metal resource, the control of which had to be of fundamental importance. It is plausible that the Antas Temple was the place where the Carthaginian authority managed the trade of rough lead and silver. If this hypothesis turns out to be well-founded, it would explain the high importance of the temple and would justify an assault with consequent destruction of the temple by mercenaries, i.e., the damage to the votives found in Antas could not be attributable to natural causes, but to a deliberate *damnatio memoriae* against Carthage that occurred at the end of the third century BC.

The subsequent period of Roman occupation is testified by numerous remains of mining works, by the tools and products obtained in the foundries and by the presence of mining settlements such as *Metalla* and Antas [6].

The area has been the subject of studies by archaeologists, historians and architects, ([1,2,4] and references therein), but no studies have been carried out on the geomaterials used in historical times. In the scientific literature there are several geological studies of local lithologies, but no correlation has ever been made between these and the construction materials found in the Antas archaeological area.

This research intends to offer a valuable contribution primarily to petrographic study, and to classify the materials used for the construction of the two temples found in the area of Antas referable to both the Punic and Roman phases (Figure 1).

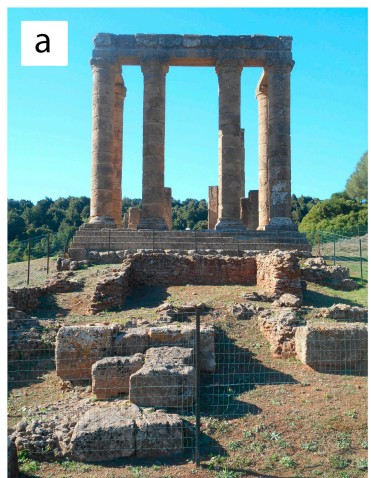 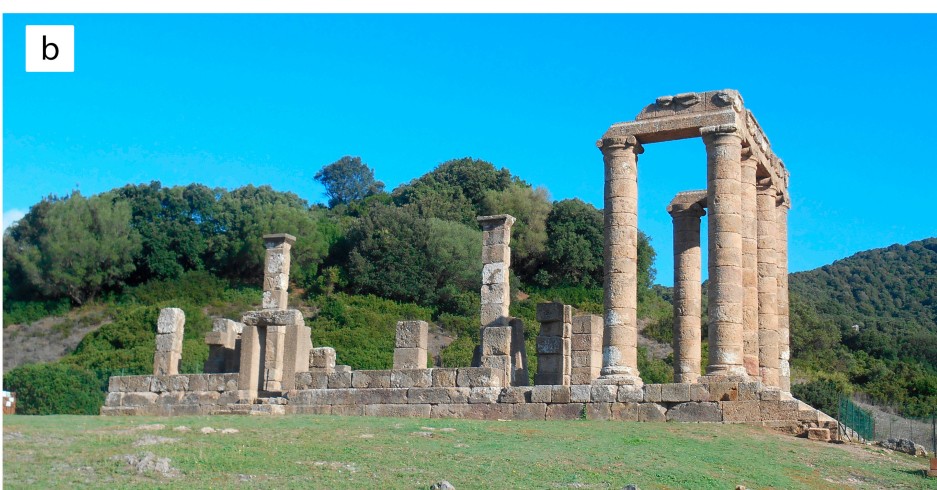

**Figure 1.** (**a**) Front view of the Punic temple; (**b**) lateral view of the Roman temple.

Studies aimed at the compositional characterization of stone construction materials can solve problems inherent to historical archaeometry [7–13], or to identify the geographical origin of the lithologies used in the monument [14–20], and, together with studies of ancient mortar [21–27], to define construction phases. The investigations on the materials also allow resolution of archaeometric-conservative issues [28–31], as to intervene in the conservation-restoration [32–34] it is certainly necessary to have a technical-scientific basis of data on the compositional aspects of the materials and their chemical, physical and mechanical properties [35–39].

Specifically, the aim of this work is the minero-petrographic and physical-mechanical characterization of stone materials to define, not only their technical-constructive characteristics and the state of conservation and their origin from the surrounding area with the precise geographical location of the historical quarries, but also to indirectly understand commercial routes in that historical period and the possible exchanges between the settle-

ments present in the Sulcis area. In fact, the Antas site is of fundamental importance from a cultural point of view as it testifies to the presence of different civilizations over time, starting from the Nuragic period (second millennium BC) to the Roman one (third century AD).

In recent times, the Superintendence of Archaeological Heritage of Cagliari and the Province has encountered various problems concerning the Temple following the restoration work that took place in the 1970s in which, in addition to a partial anastylosis of the structure and a laying of the stone ashlars found scattered around the site, an important consolidation of the reconstruction work was carried out with the massive use of artificial lithoid materials (e.g., mortars and concretes with a cement binder for the reconstruction of the missing stone ashlars and decorative elements of temple). For this reason, in addition to the natural stones used for the construction of the buildings, this work deals with the study of the artificial geomaterials used for the reconstruction of the work and the aforementioned restoration.

## 2. Historical and Archaeological Evidences

In the Middle Ages the writers L'Anonimo Ravennate (7th century) and Guidone (12th century), cited the temple of *Sardus Pater* in their geographical works written using ancient sources, placing the temple along a road between *SulKi* (Sant'Antioco) and *Neapolis* (Guspini-*Santa Maria de Nabui*) [1].

In 1580 the scholar and bishop of Sassari, Gianfrancesco Fara placed the temple of *Sardus Pater* on the "Caput Neapolis", that is, on today's Capo Pecora promontory. Thirty years later, based on the data of Ptolemy, the Dutch geographer Filippo Clüver placed the temple on Capo Frasca, the promontory delimiting the Gulf of Oristano to the South. Given that the ruins of the temple were not found, Clüver also reinterpreted Ptolemy's scripts suggesting that he had not mentioned a temple (ieròn), but a promontory (akron) dedicated to the divinity; this hypothesis was soon discarded [1].

The search for the *Sardus Pater* temple subsided in the 18th century, when various scholars concentrated exclusively on the study of the coin struck in Sardinia with the effigy of a deity, and resumed in 1825 with Giuseppe Manno, who was uncertain about the location of the temple between Capo Pecora and Capo Frasca. The latter location was also indicated in the same period by the Piarist Vittorio Angius, who suggested a second possible location at the top of Monte Arcuentu, in the Guspinese [1].

In 1838 General Lamarmora, during his journey through the island for drafting of the work "Voyage en Sardaigne" [40], arrived in the Riu Antas valley and recognized the remains of a temple and fragments of columns and capitals dispersed in the forest and covered by vegetation (Figure 2). He noticed that the base of the building was mainly intact, even if invaded by a forest of oak trees which accelerated its destruction. In 1840, following the publication (in 1839) of the temple reliefs of the architect Gaetano Cima (Figure 3) and a partial reading of the epigraph, Lamarmora attributed the temple to Antonino Pio (138–161 AD) or Marcus Aurelius (161–180 AD). In this case, the General Lamarmora identified it as a Roman construction, belonging to the class of tetrastyle temples due to the presence of four columns placed in front of the pronaos, on the sides of which there were another column and two corner pillars. The finding suggested that the temple was to be an extra-urban sanctuary in the territory of the mining town of *Metalla*, mentioned in the *Itinerarium Antonini*, located between *Neapolis* and *Sulki* [1,4,40,41]. Even if the route of the Roman road indicated in the Itinerarium cannot be accurately reconstructed, the thirty Roman miles (45 km) of the *Neapolis-Metalla* stretch and the *Metalla-Sulki* stretch lead to locate *Metalla* at Fluminimaggiore, the municipality of the Antas site. This idea seemed to be supported by the discovery, in some localities of the island, of a Roman coin from the second half of the 1st century BC, which carried on the reverse side a tetrastyle temple, identified with the Temple of Antas, and the letter M, considered the abbreviation by *Metalla* [1].

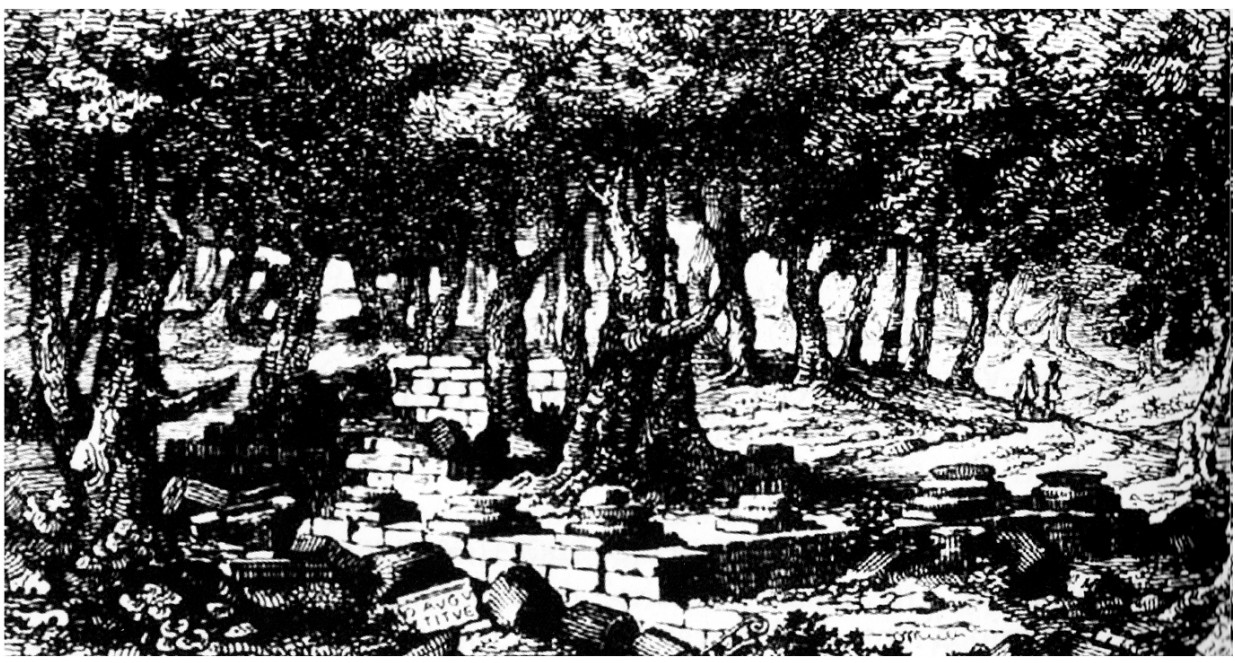

**Figure 2.** Representation of the ruins of the temple of Antas (from Lamarmora [40]).

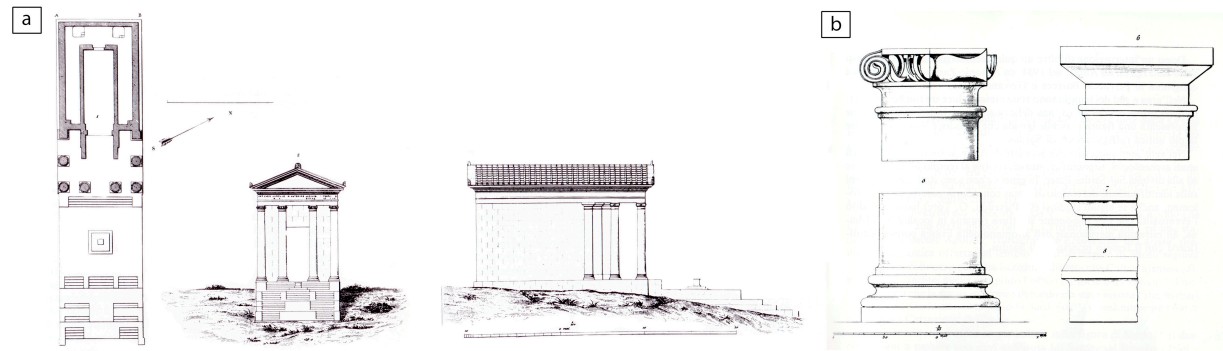

**Figure 3.** (**a**) Relief and ideal reconstruction of the temple of Antas by the architect Gaetano Cima (from Lamarmora [40]); (**b**) fragments of columns and capitals drawn by the architect Gaetano Cima (from Lamarmora [40]).

In 1858, the canon Giovanni Spano and Vincenzo Crispi carried out more detailed research but without carrying out excavations. Studies concerning the temple resumed from the late 19th century, a period in which the epigraphist Schiemt and Ettore Pais reported the inscription to the princedom of the emperor Commodus (180–192 AD) [1].

Only in the early 1960s of the 20th century, the researcher Foiso Fois, analysing two small quadrangular environments, which closed the *sacellum* on the short northwestern side, understood that the building technique used in the temple was different from that used for structures under the access steps; thus he hypothesized the Punic origin of the Temple of Antas [1].

The question of the location of the temple was resolved following an important survey carried out in 1966 in the Antas valley, where the ruins of the nuragic village and others archaeological findings, including some Roman tombs and ancient stone quarries, were discovered [1,42]. The finding of one bronze table with the dedication to *Sardus Pater* dispelled any doubts.

In the years 1967 to 1968, the Superintendence for Antiquities of Cagliari and the Institute of Near Eastern Studies of the University of Rome, led by Gennaro Pesce and Sabatino Moscati, decided to promote an excavation in the locality of Antas, directed by Ferruccio Barreca. Various building blocks and column drums, as well as various finds and

coins from the Punic and Roman ages, and decorative architectural systems, were recovered. New fragments of the architrave epigraph were found which indicated the dedication of the temple of *Sardus Pater*, and various Punic stone and bronze epigraphs, dated between the 4th and 3rd century BC, with dedications to Sid. He managed to understand with certainty that the temple of the Roman *Sardus Pater* of the imperial age, stood on the area of an older Punic place of worship dedicated to Sid [1].

In 1976 the anastylosis of the temple ended (Figure 1), directed by Ferruccio Barreca, and, after an interruption, the archaeological excavations resumed in 1984 with Antonio Zara under the joint direction of Ferruccio Barreca and Giovanni Ugas, during which three tombs of the early Iron Age were discovered about 20 m from the podium of the Roman temple [1].

In 2003 and 2004, new excavations were carried out in the area conducted by the archaeologist Michela Migaleddu, whose results are being published.

## 3. Geological Setting

The Sardinia-Corsica block (Figure 4) is a continental microplate located in the western Mediterranean between the Ligurian-Provencal Basin and the Tyrrhenian Basin. Its present location is due to Cenozoic tectonic movements leading to its separation from the main European continent (southern France and the Iberian Peninsula) [43–50]. Sardinia consists of a Variscan basement (intrusive and metamorphic rocks of variable degree, see small Sardinia sketch map in Figure 4, [51]) of Early Cambrian to Pennsylvanian age partially covered by Late Palaeozoic to Pliocene-Quaternary sedimentary and volcanic rocks [52–58]. The widest and most continuous volcano-sedimentary covers crop out along the "Sardinia Trough", an Oligo-Miocene depression that crosses the island from north to south [52–55], whose southern area was partially reactivated in the Pliocene-Quaternary forming the Campidano graben [56].

The Antas Temples is located in southwestern Sardinia (Iglesiente subregion) (Figure 4) that, from a geological point of view, pertains to the external zone of the Sardinian Variscan basement [59,60] here characterized by the widespread occurrence of polydeformed, low to very low-grade metamorphic rocks belonging to a former sedimentary succession ranging in age from the Early Cambrian to the Late Ordovician (Figure 5). Different subdivisions of the Cambrian terms are proposed by authors [61–63]. Here we adopt the subdivision reported in the new 1: Iglesias Geological Sheet 555 by the Italian CARG project ([64], see legend of Figure 5).

In the study area, the succession starts with the mainly terrigenous sediments of the Nebida Fm (Fm = Formation), made up of light grey metasiltstones (Matoppa Mb (Mb = Member)) at the bottom, and overlaid by rhythmic alternations of metasandstones, metasiltstones and metaclaystones with carbonate cement preserving algal mat laminae (Punta Manna Mb).

The Gonnesa Fm ("Gonnesa Group" ([65], and references therein)) marks the change from a mainly siliciclastic to a clean carbonate sedimentation. The lower "Laminated Metadolostone" Mb consists of alternating, well stratified metadolostones and metalimestones hosting algal mats, oolithic-oncolithic levels, thinly stratified carbonate muds and rare evaporites. The upper "Ceroid Limestone" Mb is made up by dark to light-grey, thickly stratified to massive metalimestones, locally affected by diagenetic dolomitization [66]. Eight kilometers eastward in the next Oridda area, a detailed sedimentological investigation of the carbonate Gonnesa Group was performed by Costamagna [67]: its sedimentological and paleogeographic results have been extended in the conclusions to the presently studied area.

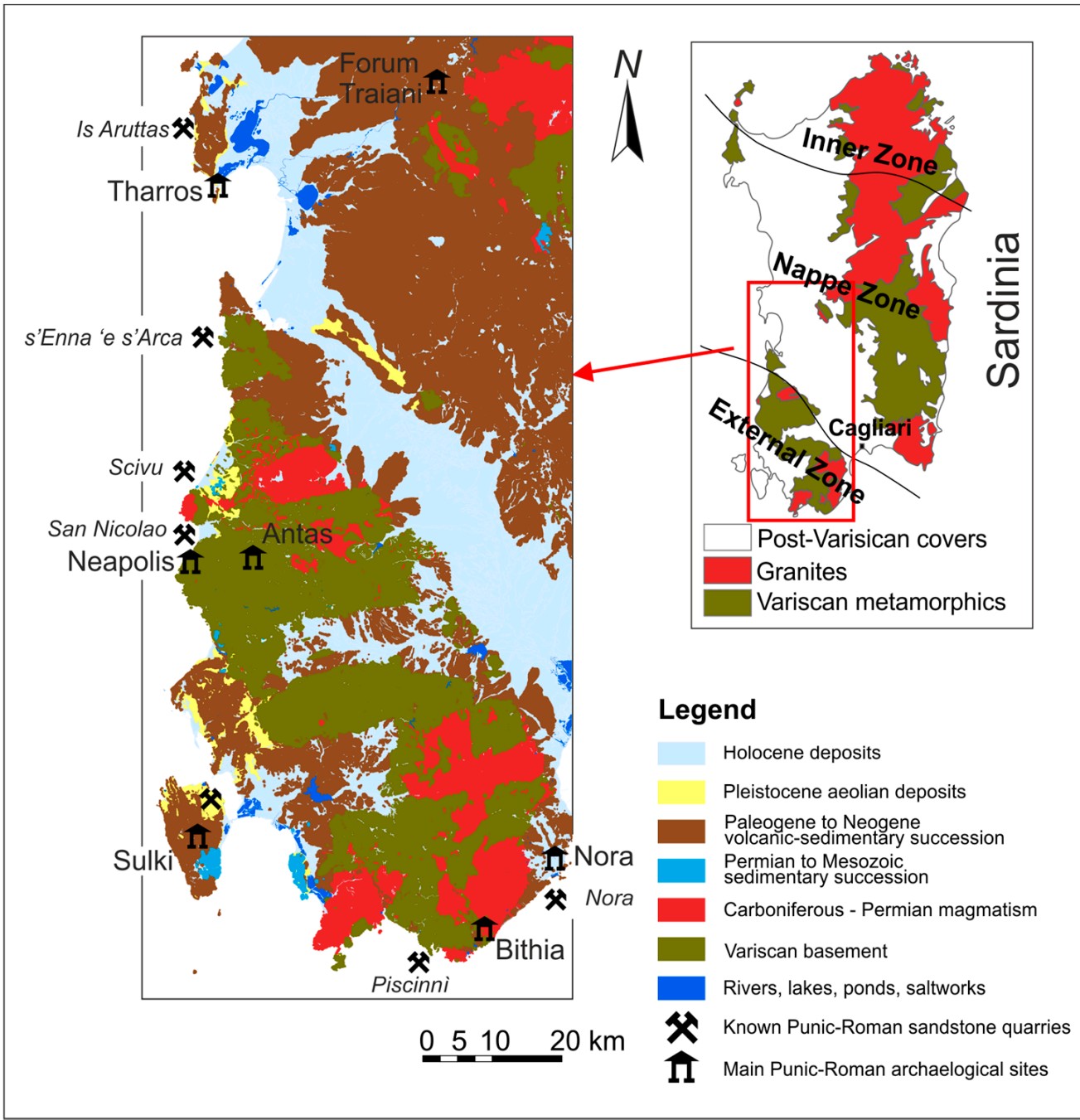

**Figure 4.** Geological sketch-map of the southwestern Sardinia coast with the location of the Antas Temple and other important archaeological sites from Punic and Roman times such as *Tharros*, *Forum Traiani*, *Neapolis* (with probable localization near to Antas site), *Sulki*, *Bithia*, *Nora*, and ancient Punic-Roman quarries, such as *Is Aruttas*, *s'Enna 'e s'Arca*, *Scivu*, *San Nicolao*, *Piscinnì*, *Fradis Minoris* and *Sa Perededda* (near to *Nora* site). Geological base from RAS, 2010, modified [51].

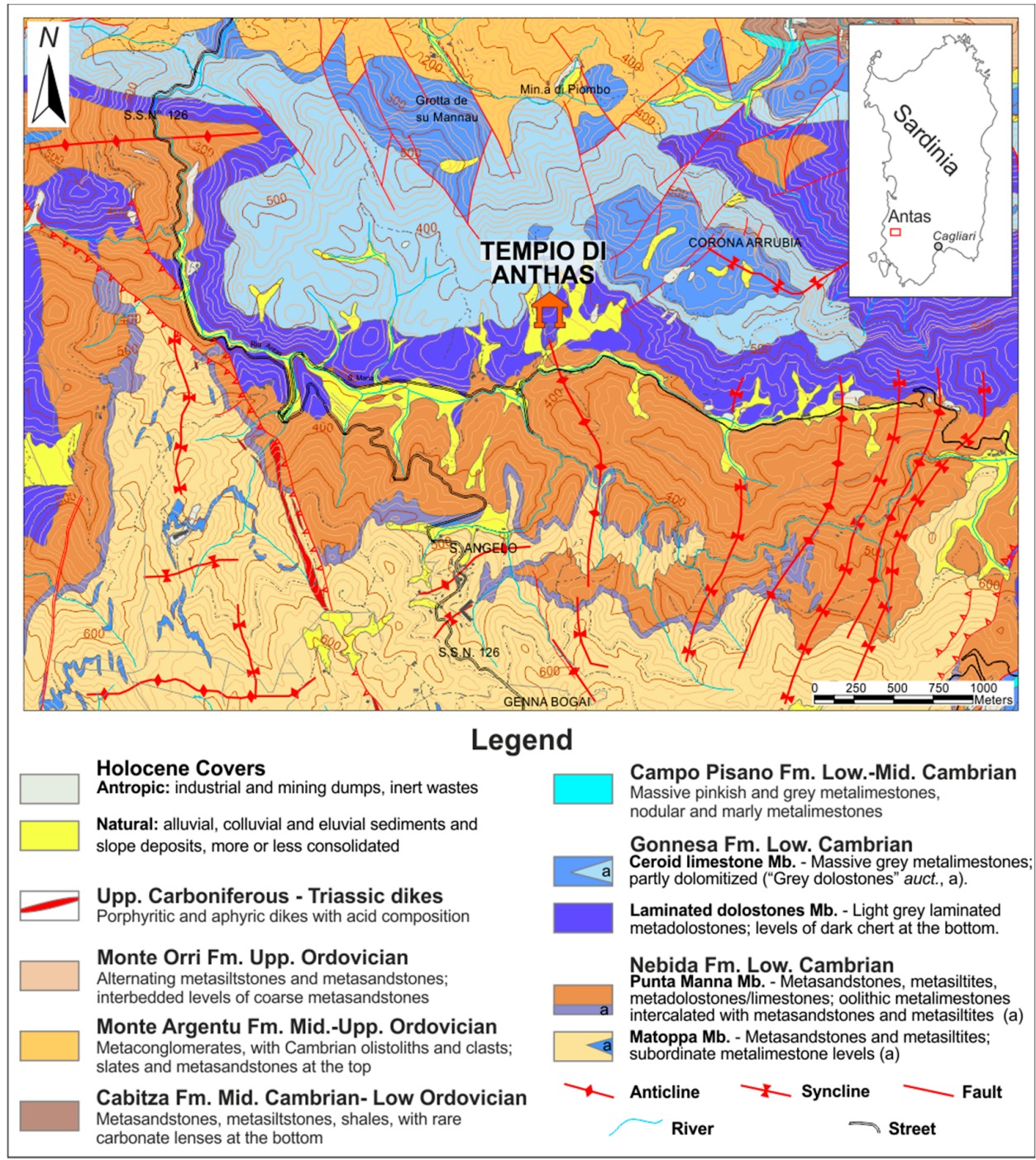

**Figure 5.** Geological map of the area surrounding the Punic-Roman archaeological site of Antas (Fluminimaggiore, Sulcis, SW Sardinia). From RAS, 2010, modified [51].

The Campo Pisano Fm is mainly formed by grey to pinkish nodular metalimestones with an abundant fossiliferous content that dates the base of the formation at the middle Cambrian. Subordinate facies, consisting of massive metalimestones and metamarls, are also found.

The Early Cambrian-Lower Ordovician succession ends with the "Cabitza Schists" Fm, consisting of a rhythmic alternating metaclaystones, metasiltstones, and fine-grained metasandstones; the upper part of the formation has been assigned biostratigraphically to the Early Ordovician.

The contact between the Cabitza Fm and the overlaying Monte Argentu Fm is erosive and markedly unconformable due to a Middle Ordovician tectonic event in a convergent geodynamic setting [68] known as the "Sardic Phase" Auct. The Monte Argentu Fm, formerly known as "Puddinga", starts with heterometric and polygenic metaconglomerates and metabreccias, metre-sized "olistoliths" embedded in reddish metasandstones and metasiltsones [69] followed by metasiltstones and slates. The gradual transition toward the overlaying marine Monte Orri Fm is marked by a change in the colour of metasediments, from reddish to olive-green/grey and by the appearance of hummocky cross stratifications (HCS, [70]) in the lower part of Monte Orri Fm [71].

The Lower Palaeozoic metasediments in the study area are cut by aphyric to porphyritic dikes of Late Pennsylvanian to Triassic age [72]. They range in composition from rhyodacite to rhyolite. Younger Holocene sedimentary covers are mainly represented by alluvial and eluvial-colluvial deposits but also by anthropic deposits, especially mine and industrial wastes.

## 4. Materials and Methods

### 4.1. Sampling

A geological and petrographic survey on the Antas monuments, the surrounding archaeological site, including the ancient quarries and natural outcrops, was carried out after an accurate reading of architectural, archaeological and geological literature about the Antas site.

A systematic sampling of stonework, ashlars and outcropping rock (Figures 6–9) was planned, taking into account the architectural features of the site (planimetry, building technologies, wall textures, etc.) and the macroscopic lithological characteristics of geomaterials, including their lithology, decay phenomena and conservation state. The sampling was performed according to the Recommendations Nor.Ma.L. 3/80 [73] and to the limitation of the local Superintendence of Cultural Heritage, which imposes strict limits on the quantity of samples collected. Nevertheless, the amount of each sample was reliably considered representative and adequate for the analytical techniques. Each sampling point was georeferenced and reported on the site plan maps and photographed. The corresponding samples were catalogued and documented by a preliminary macroscopic description and by macro and microphotographs (see Table 1). Each sample was cut to obtain specimens for physical and mechanical tests and to prepare polished thin sections; an aliquot was finely ground for mineralogical and physical analyses.

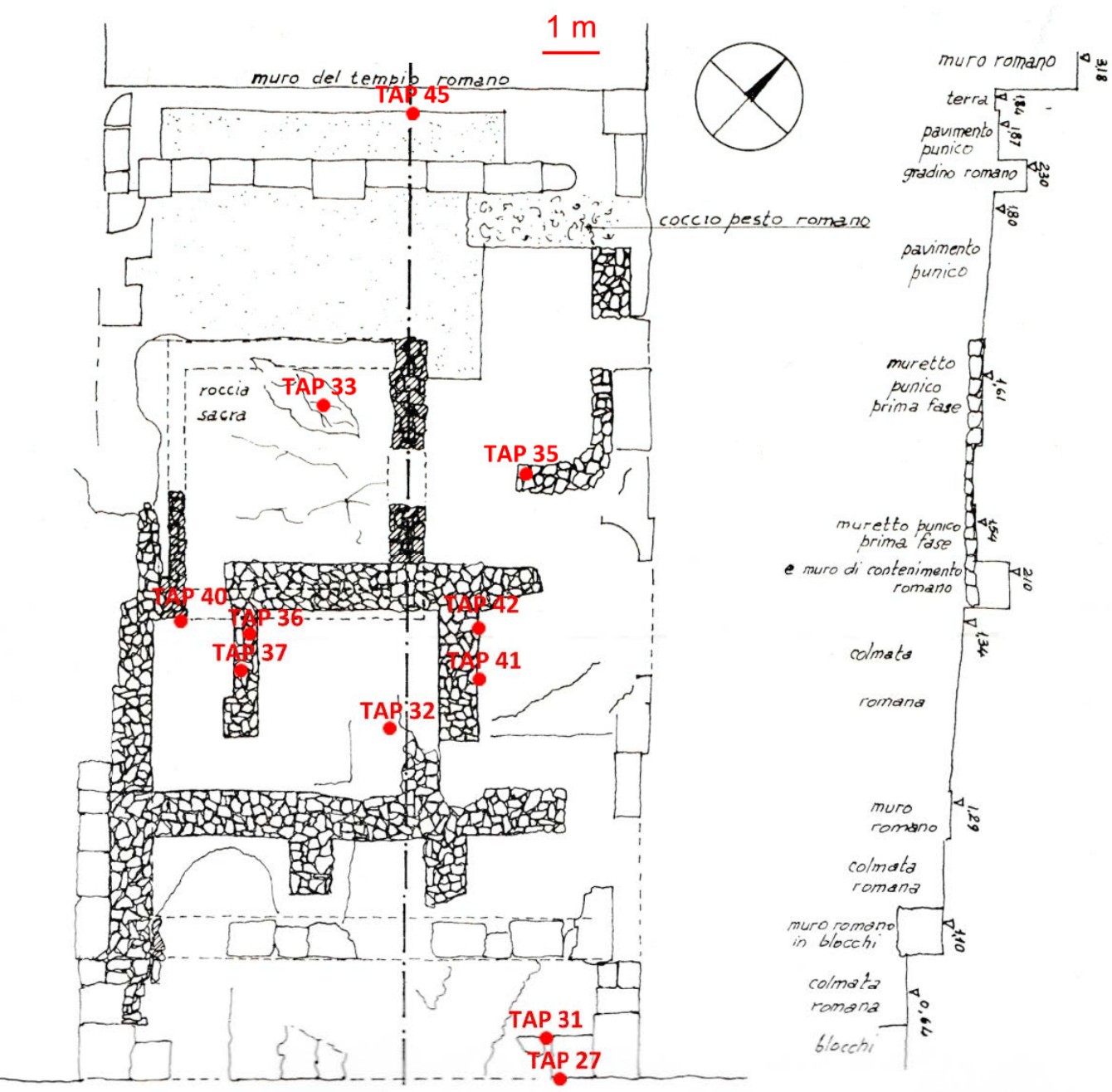

**Figure 6.** Sampling position on the map from the Punic Temple (from Zucca [1], modified).

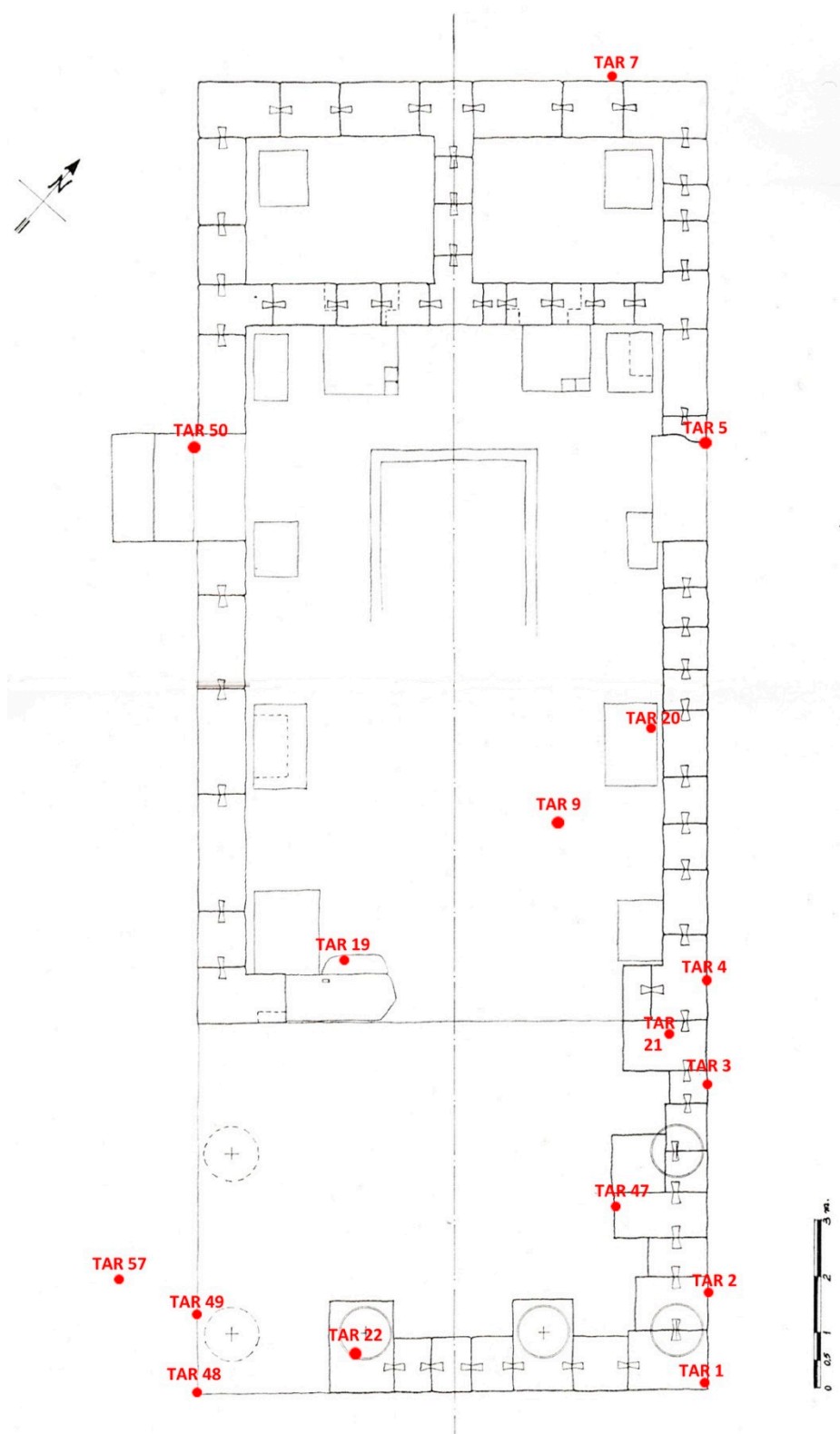

**Figure 7.** Sampling position on the map from the Roman Temple (from Zucca [1], modified).

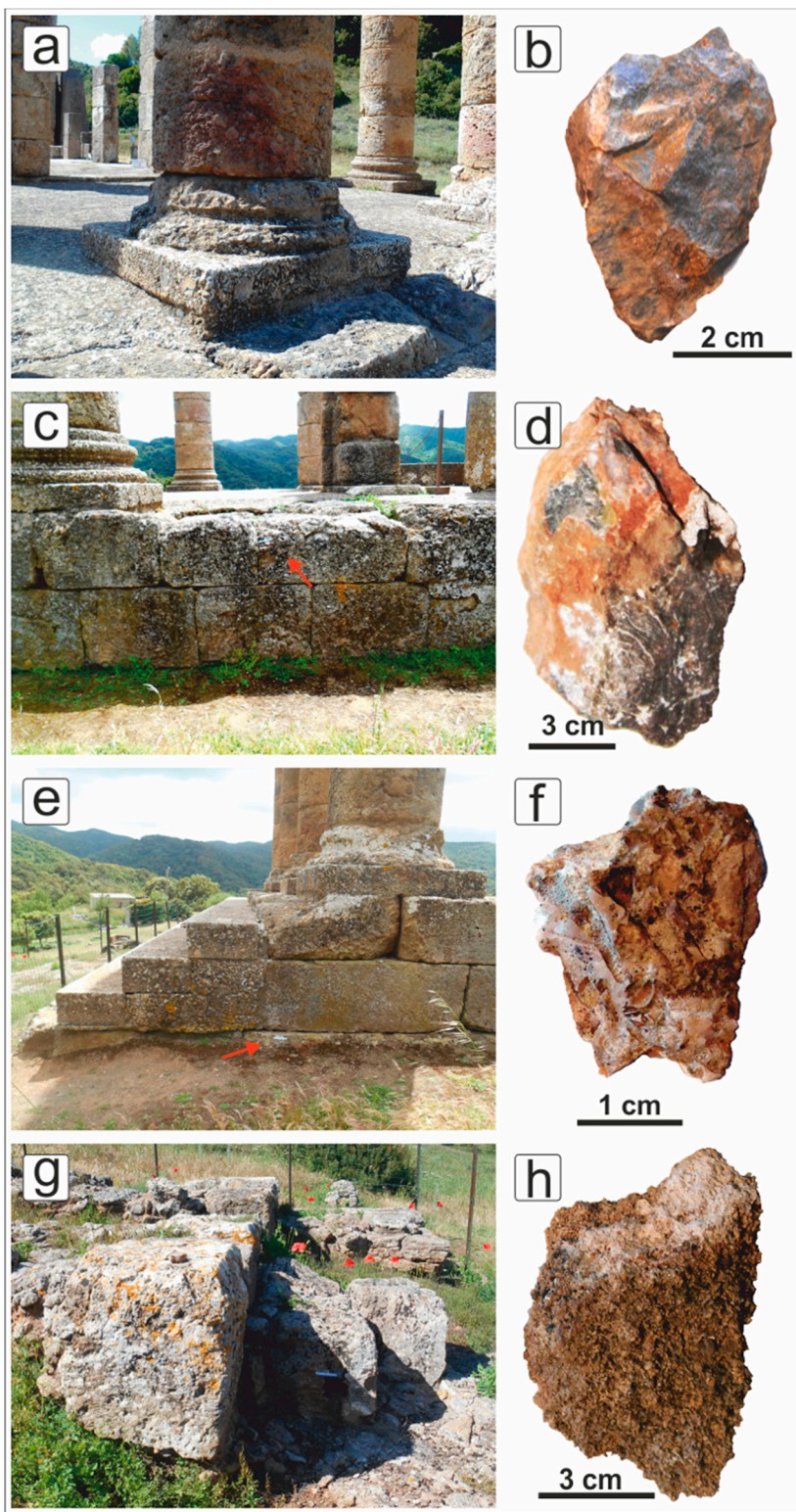

**Figure 8.** (**a**,**b**) massive metadolostone sampled in the Roman temple (sample TAR22); (**c**,**d**) laminated metadolostone sampled in the Roman temple (sample TAR3); (**e**,**f**) brecciated metadolostone collected at the base of Roman temple (sample TAR1); (**g**,**h**) fossiliferous sandstone (samples TAP23, 27, 31) from the Punic temple. Note: the red arrows indicate the sampling points.

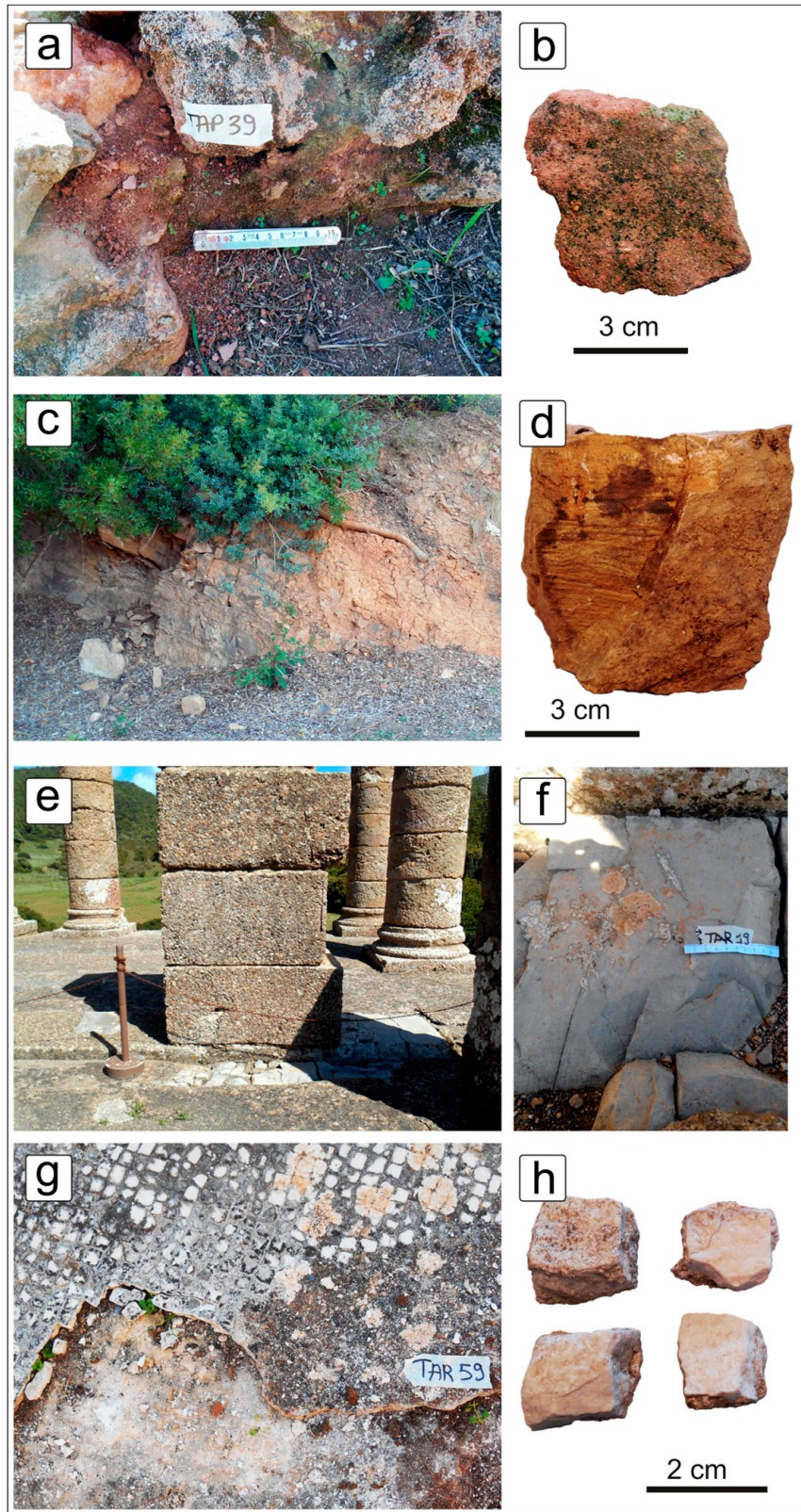

**Figure 9.** (**a**,**b**) Sandstone-like rock ("Sandstone" *s.l.*) from the Punic temple; (**c**,**d**) metasiltstone (sample TASC7) collected along the path to the ancient Roman quarries (samples TAC1, TAC2, TAC3); (**e**,**f**) marly limestone (TAR19) sampled in the *Pronao* of the Roman temple; (**g**,**h**) limestone tesserae (sample TAR9) from the mosaic of the Roman temple.

**Table 1.** Sample list summarizes the sampling position of stones and lithology description. "Sandstone" *s.l.* = "Sandstone" *sensu lato*.

| Sample | Monument | Sampling Position | Lithology | Description of Rock |
|---|---|---|---|---|
| TAR1 | Roman Temple | Access steps | Metadolostone | Weathered massive metadolostone |
| TAR2 | Roman Temple | Access steps | Metadolostone | Dolomitic metabreccia with calcite cement |
| TAR3 | Roman Temple | Perimeter of the podium | Metadolostone | Peloidal metadolostone showing voids filled by calcite |
| TAR4 | Roman Temple | Perimeter of the podium | Metadolostone | Dolomitic metabreccia with calcite cement |
| TAR5 | Roman Temple | North-east side entrance | Metadolostone | Crystalline laminated metadolostone with ghost of pre-existing structures |
| TAR7 | Roman Temple | Perimeter of the podium | Metadolostone | Dolomitic metabreccia with recent sparry calcite cement |
| TAR9 | Roman Temple | Mosaic tesserae of cell floor | Limestone | Mesozoic bioclastic packstone-grainstone |
| TAR14 | Roman Temple | North-western basin | Sandstone | Calcarenaceous sandstone with fossil fragments and carbonate cement |
| TAR19 | Roman Temple | Floor near the entrance | Marly limestone | Laminated, clayey mudstone-wackestone with tiny planktonic forams (Mesozoic) |
| TAR20 | Roman Temple | Pillar of cell-room | Metadolostone | Massive metadolostone with ghosts of pellettoidal texture |
| TAR22 | Roman Temple | Column base | Metadolostone | Massive metadolostone with ghosts of pellettoidal texture |
| TAR49 | Roman Temple | Perimeter of the podium | Metadolostone | Massive metadolostone with ghosts of pellettoidal texture |
| TAR50 | Roman Temple | Steps of lateral access | Sandstone | Calcarenaceous sandstone with bioclasts and carbonate cement |
| TAP23 | Punic Temple | Stone ashlar of masonry | Sandstone | Calcarenaceous sandstone with bioclasts and carbonate cement |
| TAP27 | Punic Temple | Stone ashlar of masonry | Sandstone | Calcarenaceous sandstone with bioclasts and carbonate cement |
| TAP30 | Punic Temple | Stone ashlar of masonry | Sandstone-like rock | "Sandstone" *s.l.* with quartz, rare feldspars and lithics |
| TAP31 | Punic Temple | Stone ashlar of masonry | Sandstone | Calcarenaceous sandstone with bioclasts and carbonate cement |
| TAP32 | Punic Temple | Stone ashlar of masonry | Sandstone-like rock | "Sandstone" *s.l.* with quartz, rare feldspars and lithics |
| TAP33 | Punic Temple | Outcrop (sacred Temple) | Metadolostone | Dolomitized algal bindstone |
| TAP35 | Punic Temple | Stone ashlar of masonry | Sandstone | Calcarenaceous sandstone with bioclasts and carbonate cement |
| TAP36 | Punic Temple | Stone ashlar of masonry | Sandstone | Calcarenaceous sandstone with bioclasts and carbonate cement |
| TAP37 | Punic Temple | Stone ashlar of masonry | Sandstone-like rock | "Sandstone" *s.l.* with quartz, rare feldspars and lithics |
| TAP38 | Punic Temple | Stone ashlar of masonry | Sandstone-like rock | "Sandstone" *s.l.* with quartz, rare feldspars and lithics |
| TAP39 | Punic Temple | Stone ashlar of masonry | Sandstone-like rock | "Sandstone" *s.l.* with quartz, rare feldspars and lithics |
| TAP40 | Punic Temple | Stone ashlar of masonry | Sandstone-like rock | "Sandstone" *s.l.* with quartz, rare feldspars and lithics |
| TAP41 | Punic Temple | Stone ashlar of masonry | Sandstone-like rock | "Sandstone" *s.l.* with quartz, rare feldspars and lithics |
| TAP42 | Punic Temple | Stone ashlar base | Sandstone | Calcarenaceous sandstone with bioclasts and carbonate cement |
| TAP45 | Punic Temple | Ashlar wall (in front steps) | Metadolostone | Weakly brecciated, weathered metadolostone |
| TAP48 | Punic Temple | Stone ashlar not in situ | Sandstone | Calcarenaceous sandstone with bioclasts and carbonate cement |

**Table 1.** *Cont.*

| Sample | Monument | Sampling Position | Lithology | Description of Rock |
|--------|----------|-------------------|-----------|---------------------|
| TAC1 | Roman quarry | Worked rock outcrop | Metadolostone | Dolomitic breccia with massive metadolostone angular pebbles |
| TAC2 | Roman quarry | Worked rock outcrop | Metadolostone | Dolomitized algal bindstone |
| TAC3 | Roman quarry | Worked rock outcrop | Metadolostone | Dolomitic breccia with massive metadolostone angular pebbles |
| TASC1 | Rock outcrop | Path temple-quarries | Metadolostone | Massive metadolostone with calcite-filled fractures |
| TASC2 | Rock outcrop | Path temple-quarries | Metadolostone | Massive metadolostone with thin levels of microbreccias |
| TASC3 | Rock outcrop | Path temple-quarries | Metadolostone | Laminated metadolostone with algal mats and peloids |
| TASC4 | Rock outcrop | Path temple-quarries | Metadolostone | Laminated metadolostone with algal mats and peloids |
| TASC5 | Rock outcrop | Path temple-quarries | Metadolostone | Crushed laminated metadolostone with peloids |
| TASC6 | Rock outcrop | Path temple-quarries | Metadolostone | Peloidal metadolostone showing several overprinted stages of dolomitization |
| TASC7 | Rock outcrop | Path temple-quarries | Metasiltstone | Laminated fine litharenite (Fm di Nebida, Lower Cambrian?) |
| TASC8 | Rock outcrop | Path temple-quarries | Metadolostone | Metadolostone with rare laminae |

### 4.2. Specimens and Analytical Methods

Polished thin sections, about 30 μm thick, were made for analysis by optical microscopy RPL-3T, Radical (Haryana, India). Sandstone classification was made by qualitative evaluations using comparing atlases and table [74] and then following the QFR method [75,76]. Carbonate matrix-rich sandstones have been sorted and classified by identifying the remaining terrigenous grains but also considering conveniently the Mount's classification [77] when the carbonate intraclastic part was significant. Clean carbonate rocks were classified by using the Dunham scheme [78] expanded by Embry and Klovan [79], and Flugel's tables [80].

Cubic specimens (with an average size of 15 mm × 15 mm × 15 mm) were cut by a diamond wheel for determining their physical and mechanical properties. Physical tests with the measurement of real and bulk volumes, and dry and wet masses were carried out according to Buosi [33] and Columbu et al. [81,82]. The specimens were dried at $105 \pm 5\,°C$ and the dry mass ($m_D$) was determined. The real volume ($V_R$) with: $V_R = V_S + V_C$ (where: $V_S$ is the volume of solid phases, $V_C$ is the volume of pores closed to helium of the specimens) was determined by a helium Ultrapycnometer 1000 (Quantachrome Instruments, Boynton Beach, Florida). Then, the wet solid mass ($m_W$) of the samples was determined until constant weight. Through a hydrostatic analytical balance, the bulk volume ($V_B$) with $V_B = V_S + V_O + V_C$, where $V_O = (V_B - V_R)$ is the volume of open pores to helium and $V_B$, was calculated as:

$$VB = \left[ \frac{(m_w - m_{HY})}{\rho_{WT}} \right] \cdot 100 \tag{1}$$

where $m_{HY}$ is the hydrostatic mass of the wet specimen and $\rho_W$ T (25 °C) is the water density at a temperature of 25 °C.

Total porosity ($\Phi_T$), water/helium open porosity ($\Phi_O$ $H_2O$-He), water/helium closed porosity ($\Phi_C$ $H_2O$-He), weight imbibition coefficient ($IC_W$), saturation index (SI), real ($\rho_R$) and bulk($\rho_B$) were computed as:

$$\Phi_T = \left[ \frac{(V_B - V_S)}{V_B} \right] \cdot 100 \tag{2}$$

$$\Phi_O H_2O = \left\{ \frac{\left[ \frac{(m_W - m_D)}{\rho_{WTx}} \right]}{V_B} \right\} \cdot 100 \tag{3}$$

$$\Phi_O He = \left[ \frac{(V_B - V_R)}{V_B} \right] \cdot 100 \tag{4}$$

$$\Phi_C H_2O = \Phi_O He - \Phi_O H_2O \tag{5}$$

$$\Phi_C He = \Phi_T - \Phi_O H_e \tag{6}$$

$$IC_w = \left[ \frac{(m_w - m_D)}{m_D} \right] \cdot 100 \tag{7}$$

$$SI = \left( \frac{\Phi_O H_2O}{\Phi_O H_e} \right) = \left\{ \frac{\left[ \frac{(m_W - m_D)}{\rho_{WTx}} \right]}{V_O} \right\} \cdot 100 \tag{8}$$

$$\rho_R = \frac{m_D}{V_R} \tag{9}$$

$$\rho_B = \frac{m_D}{V_B} \tag{10}$$

The strength index (PLT index) was determined with a Point Load Tester (mod. D550 Controls Instrument, Milan, Italy), according to ISRM (International Society for Rock Mechanics and Rock Engineering) Recommendations [83,84]. The load was applied via the application of a concentrated load with two opposing conical punches. The strength index (Is) was calculated as $Is = \frac{P}{De^2}$ where P is breaking load and De the "equivalent diameter of the cylindrical specimen", with $De = \frac{4A}{\pi}$; $A = W \cdot D$ where W and 2 L are the width perpendicular to the load direction and length of specimen, respectively, and D is the distance between the two conical punches. The index value was referred to a standard cylindrical specimen with diameter D = 50 mm for which Is was corrected with a shape coefficient (F) and calculated as $Is_{(50)} = Is \cdot F = Is \cdot \left( \frac{De}{50} \right)^{0,45}$.

## 5. Results

### 5.1. Petrographic Characteristics

The petrographic analyses of the thin sections allowed recognition of the different lithologies employed in the building of the Punic and Roman temples and to make a preliminary comparison with the rocks outcropping in the study area. In the following description, the analysed samples are grouped on the basis of their lithology. The sample list is reported in Table 1.

#### 5.1.1. Metadolostone

Most of the collected samples (TAR1–5, 7, 20, 22, 49, TAP33, 45, TAC1–3, TASC1–6, 8; Table 1) were metadolostones that represented the main lithology outcropping around the archaeological area together with "Ceroide Limestone" Mb (not used as a building material).

Metadolostone samples occur in different facies that can be subdivided in three main groups namely laminated, massive and brecciated dolomites.

Massive metadolostone (TAR1, 20, 22, 49, TASC1–2; Table 1; Figure 10a) was characterized by the absence of sedimentary structures even if some samples still preserved relics of peloid-rich beds and desiccation fenestrae. The presence of metadolostone samples with intermediate features between laminated and massive, suggested that at least some of the latter may have resulted from a progressive secondary dolomitization/recrystallization that obliterate primary structures. Nonetheless, massive dolomitized carbonate muds completely devoid of structures may exist too. Here millimeter-sized dolomite rhombohedra could be found locally.

Laminated metadolostone (TAR3, TAR5, TAP33, TAC2, TASC3–6, TASC8; Table 1; Figure 10b) were characterized by the occurrence of thin algal mats ("stromatolites" AA) and peloid/pellets or oncoidal-oncolitic-rich beds. Interparticle voids, filled by calcite (fenestral fabric: [80,85]) were quite common and are probably linked to desiccation cycles, in a sabkha-like environment.

Brecciated metadolostone (TAR2, 4, 7, TAC1, 3, TAP45; Table 1; Figure 10c), either massive or layered, were characterized by a millimeter to centimeter-sized network of randomly distributed fractures that separate angular metadolostone clasts heterogeneous in size. Fractures were commonly filled by dolomite and/or calcite cement with variable amounts of iron oxides. Syn-sedimentary and tectonic recent breccias can be distinguished from each other on the basis of the isoorientation and the good sorting of the synsedimentary elements and the strongly angular shape and the poor sorting of the tectonically originated fragments.

### 5.1.2. Sandstone

Almost all sandstone samples were collected from the Punic temple (TAP23, 27, 31, 35, 36, 42, 48) except two of them, belonging to the Roman temple (TAR14, 50) (Table 1). Natural outcrops of this kind of rock were not found in the surrounding area; thus it was not possible to collect in-situ samples. Samples of this group were matrix-poor hybrid arenites *sensu Zuffa* [86] (calcarenaceous sandstones, [87]; sandy allochem limestones *sensu* Mount, [77]) bound by a carbonatic cement (Figure 10d). Grains mainly consisted of quartz, feldspars, carbonates and abundant fossils remains such as echinoid radiola, shell fragments and algae. Rare but always present were lithoclasts mainly of metamorphic origin. Grains were subrounded to rounded with a sorting more or less bimodal, made up by coarser grains up to 2–3 mm and finer ones of about 0.5 mm. The fossil assemblage suggested a Cenozoic (Miocene ?) age for those hybrid sandstones.

### 5.1.3. "Sandstone" *sensu lato* (*s.l.*)

Samples TAP30, 33, 37–41 (Table 1), quite different if compared to the above-described sandstones, were separately grouped. They consisted of quartz-rich sandstone with a consolidated matrix locally transformed in a grey-reddish carbonate cement (Figure 10e). Grains, mainly quartz plus subordinated feldspars and lithics were commonly well rounded and homogeneous in size (0.2–0.4 mm).

All "sandstone" *s.l.* samples were collected in the Punic Temple and, as for fossiliferous sandstones, no outcrops were found in the surrounding area. On the basis of the observed petrographic features, it is possible to hypothesize that they are not natural rocks, but an artificial sandy-conglomeratic stone looking like a sandstone, probably prepared in modern times to replace some original sandstone ashlar.

### 5.1.4. Metasiltstone

The sample TASC7 (Table 1) collected along the path to the quarries was quite different from all other samples. It was a metasiltstone with well-developed laminations marked by light-brown clay-rich levels (Figure 10f). Fine grained lithics and quartz grains were angular to subrounded, whereas muscovite grains are elongated parallel to the clay levels and show a clear crenulation. The crenulation and the isorientation of the grains reveal tectonic stress and its pertinence to the close Variscan basement. This rock is likely to belong to the Lower Cambrian "Punta Manna" Mb of the Nebida Fm, which, in its upper part, exhibits alternating siltstone and sandstones levels.

### 5.1.5. Marly Limestone

A sample collected in the Pronao of the Roman Temple (TAR19; Table 1, Figure 4) was a marly limestone not found elsewhere. It consisted of a poorly laminated mudstone-wackestone bearing different planktonic forams and indeterminable thin fragments of shells, probably belonging to pelagic bivalves (Figure 10g). Clay content seemed to be quite

low and the dark greyish colour suggested the presence of organic matter. Both the age and the provenance of this lithology was not undoubtedly determined, but it was supposed to be of the Mesozoic age and of possible non-Sardinian provenance. According to our knowledge and investigations there is no such lithotype outcropping in southern Sardinia. Even a northern Sardinia provenance is doubtful.

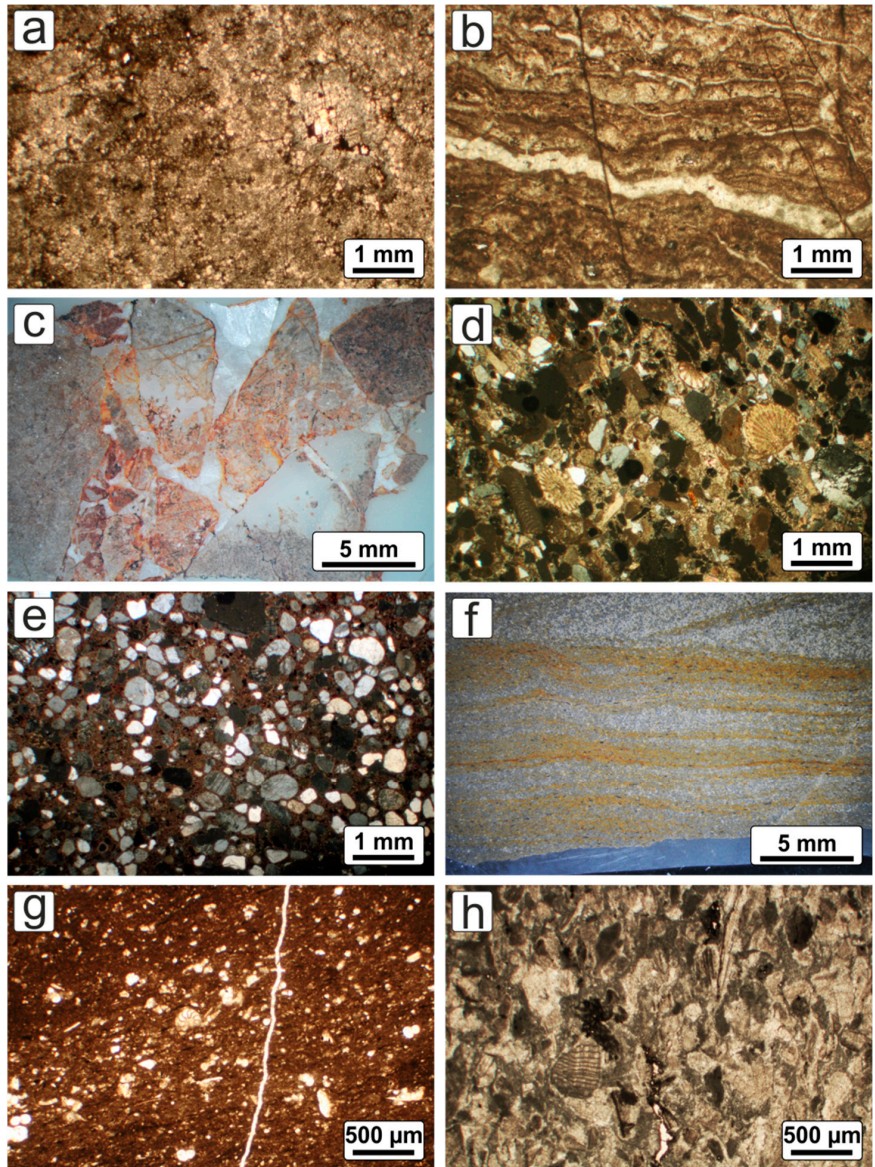

**Figure 10.** Photomicrographs of the recognized lithologies: (**a**) massive metadolostone lacking defined sedimentary fabrics (sample TAR5, 1.25x, CPL = Cross Polarized Light); (**b**) laminated metadolostone (bindstone) showing well preserved algal mats (TAR32, 1.25x, CPL); (**c**) brecciated metadolostone with angular clasts and fractures filled by calcite (TAR7, 0.63x, SMRL = Stereoscopic Microscope Reflected Light); (**d**) hybrid arenite *sensu* Zuffa [86] (sandy allochem limestone *sensu* Mount, [77]) made up by quartz, feldspars and abundant fossils in a carbonate cement (TAP27, 1.25x, CPL); (**e**) "sandstones" *s.l.* consisting of rounded submillimeter-sized quartz and feldspar grains floating in a reddish carbonate binder (TAP37, 1.25x, CPL); (**f**) siltite showing a well preserved wavy cross-stratification (TASC7, 0.63x, SMRL); (**g**) marly limestone with planktonic fauna (TAR19, 2.5x, PPL = Plane Polarized Light); (**h**) bioclastic limestone (packstone) of possible Mesozoic age exhibiting a rich fossiliferous content (TAR19, 2.5x, PPL).

### 5.1.6. Limestone

Four samples of tesserae (e.g., TAR9, Table 1) were collected from the mosaic that adorns the floor of the Roman temple. All samples were white limestones (Figure 10h) lacking any trace of deformation or metamorphism and thus considered post-Variscan. They were bioclastic calcarenite with no or few mud (packstone-grainstone). Bioclasts are hard to determine but fragments of brachiopods, crinoids, mollusc shells and possible bryozoa could be distinguished. Rare limestone and muddy lithoclasts were also found. Similarity with the limestone Lower Jurassic units of the Porto Pino Ridge located 40 km S-ward (Sulcis, [88]) was observed.

### 5.2. Petrophysical Properties

The following physical and mechanical properties were determined: real and bulk density, porosity open to helium and water, porosity closed to water, wetting coefficient by weight, water saturation index, PLT strength, compressive and tensile strengths (theoretical parameters calculated indirectly by Point Load Test data). Physical and mechanical tests were performed in all but one lithologies; indeed, the limestones used as mosaic tesserae in the Roman temple were not analysed since their scarcity did not allow performance of laboratory tests. In addition to rock samples from the monuments, outcropping metadolostone from the Roman quarries were tested. Since the sandstone outcrop was not precisely identified, it was not possible to perform tests on this material. All the data are reported in Table 2.

**Table 2.** Main following physical properties (real ($\rho_R$) and bulk ($\rho_B$) density, compact index (C), helium ($\Phi$ $H_2O$) and water ($\Phi$ He) open porosity, water closed porosity ($\Phi_C$ $H_2O$), weight imbibition coefficient ($IC_W$), water saturation index (SI)) of samples studied from the Punic and Roman temples of Antas and from outcrops.

| Sample | Lithology | $\rho_R$ | $\rho_B$ | C | $\Phi_O$ He | $\Phi_O$ $H_2O$ | $\Phi_C$ $H_2O$ | $IC_w$ | SI |
|---|---|---|---|---|---|---|---|---|---|
| | | g/cm$^3$ | g/cm$^3$ | / | % | % | % | % | % |
| TAP33 | Laminated metadolostone (from temples) | 2.80 | 2.71 | 0.97 | 3.1 | 2.7 | 0.4 | 1.0 | 86.1 |
| TAR3 | | 2.71 | 2.26 | 0.83 | 16.8 | 15.6 | 1.2 | 6.9 | 92.8 |
| TAR5 | | 2.69 | 2.17 | 0.81 | 19.5 | 17.1 | 2.4 | 7.9 | 87.9 |
| TAC2 | Lam. metadolostone (quarry) | 2.79 | 2.56 | 0.92 | 8.2 | 7.6 | 0.6 | 3.0 | 93.0 |
| TASC3 | Laminated metadolostone (from outcrops) | 2.81 | 2.69 | 0.96 | 4.3 | 4.2 | 0.1 | 1.6 | 97.7 |
| TASC4 | | 2.81 | 2.66 | 0.95 | 5.3 | 4.6 | 0.7 | 1.7 | 87.0 |
| TASC5 | | 2.82 | 2.62 | 0.93 | 7.0 | 6.4 | 0.6 | 2.4 | 91.7 |
| TASC6 | | 2.82 | 2.68 | 0.95 | 5.3 | 5.0 | 0.2 | 1.9 | 95.9 |
| TASC8 | | 2.82 | 2.65 | 0.94 | 6.3 | 6.3 | 0.0 | 2.4 | 99.5 |
| | **Average** | **2.79** | **2.56** | 0.92 | **8.4** | **7.7** | **0.7** | **3.2** | **92.4** |
| | **St. Dev.** | **0.05** | **0.20** | **0.06** | **5.7** | **5.1** | **0.7** | **2.5** | **4.8** |
| TAP45 | Brecciated metadolostone (from temples) | 2.72 | 2.40 | 0.88 | 11.7 | 11.4 | 0.4 | 4.7 | 97.0 |
| TAR2 | | 2.77 | 2.35 | 0.85 | 15.2 | 12.0 | 3.2 | 5.1 | 78.9 |
| TAR4 | | 2.80 | 2.48 | 0.88 | 11.6 | 11.2 | 0.4 | 4.5 | 96.3 |
| TAR7 | | 2.78 | 2.29 | 0.82 | 17.8 | 16.3 | 1.5 | 7.1 | 91.7 |
| TAC1 | Brecciated metadolostone (from quarry) | 2.70 | 2.48 | 0.92 | 8.5 | 6.2 | 2.3 | 2.5 | 72.5 |
| TAC3 | | 2.72 | 2.38 | 0.88 | 12.4 | 10.5 | 1.9 | 4.4 | 84.7 |
| | **Average** | **2.75** | **2.40** | 0.87 | **12.9** | **11.3** | **1.6** | **4.7** | **86.9** |
| | **St. Dev.** | **0.04** | **0.07** | **0.03** | **3.2** | **3.3** | **1.1** | **1.5** | **9.9** |
| TAR1 | Massive metadolostone (from Roman temple) | 2.74 | 2.34 | 0.85 | 14.6 | 11.6 | 3.0 | 5.0 | 79.5 |
| TAR20 | | 2.80 | 2.47 | 0.88 | 11.7 | 11.2 | 0.5 | 4.5 | 95.8 |
| TAR22 | | 2.79 | 2.55 | 0.91 | 8.7 | 6.5 | 2.2 | 2.6 | 75.1 |
| TAR49 | | 2.74 | 2.46 | 0.90 | 10.1 | 9.6 | 0.5 | 3.9 | 94.9 |
| TASC1 | Massive metadolostone (from outcrops) | 2.71 | 2.30 | 0.85 | 15.0 | 13.4 | 1.6 | 5.8 | 89.1 |
| TASC2 | | 2.79 | 2.52 | 0.90 | 9.7 | 9.1 | 0.5 | 3.6 | 94.3 |

**Table 2.** *Cont.*

| Sample | Lithology | $\rho_R$ | $\rho_B$ | C | $\Phi_O$ He | $\Phi_O$ H$_2$O | $\Phi_C$ H$_2$O | IC$_w$ | SI |
|--------|-----------|----------|----------|---|-------------|-----------------|-----------------|--------|-----|
| | | g/cm$^3$ | g/cm$^3$ | / | % | % | % | % | % |
| | **Average** | **2.76** | **2.44** | **0.88** | **11.6** | **10.2** | **1.4** | **4.2** | **88.1** |
| | **St. Dev.** | **0.04** | **0.10** | **0.03** | **2.7** | **2.4** | **1.1** | **1.1** | **8.8** |
| TAP23 | | 2.74 | 1.84 | 0.67 | 32.9 | 26.6 | 6.3 | 14.5 | 80.9 |
| TAP27 | | 2.73 | 1.96 | 0.72 | 27.9 | 22.9 | 5.0 | 11.6 | 82.2 |
| TAP31 | | 2.73 | 1.70 | 0.62 | 38.0 | 31.8 | 6.2 | 18.7 | 83.7 |
| TAP35 | | 2.70 | 2.02 | 0.75 | 25.4 | 18.2 | 7.2 | 9.0 | 71.6 |
| TAP36 | Sandstone (from temples) | 2.74 | 1.94 | 0.71 | 29.2 | 21.1 | 8.1 | 10.8 | 72.4 |
| TAP42 | | 2.73 | 1.68 | 0.62 | 38.3 | 29.9 | 8.4 | 17.7 | 78.1 |
| TAP48 | | 2.70 | 1.77 | 0.66 | 34.4 | 29.6 | 4.8 | 16.6 | 86.1 |
| TAR14 | | 2.73 | 1.71 | 0.63 | 37.4 | 35.0 | 2.4 | 20.4 | 93.5 |
| TAR50 | | 2.72 | 1.69 | 0.62 | 38.1 | 33.4 | 4.7 | 19.8 | 87.8 |
| | **Average** | **2.73** | **1.81** | **0.66** | **33.5** | **27.6** | **5.9** | **15.5** | **81.8** |
| | **St. Dev.** | **0.01** | **0.13** | **0.05** | **5.0** | **5.8** | **1.9** | **4.2** | **7.1** |
| TAP30 | | 2.60 | 1.91 | 0.73 | 26.6 | 21.3 | 5.3 | 11.1 | 80.1 |
| TAP32 | | 2.58 | 1.92 | 0.75 | 25.3 | 21.5 | 3.8 | 11.1 | 84.8 |
| TAP37 | | 2.50 | 1.98 | 0.79 | 21.0 | 16.6 | 4.4 | 8.4 | 79.2 |
| TAP38 | Artificial "sandstone" s.l. (from Punic temple) | 2.60 | 1.91 | 0.73 | 26.6 | 22.1 | 4.5 | 11.6 | 83.1 |
| TAP39 | | 2.57 | 1.96 | 0.76 | 23.7 | 17.8 | 5.9 | 9.0 | 75.2 |
| TAP40 | | 2.53 | 1.96 | 0.77 | 22.7 | 19.0 | 3.7 | 9.7 | 83.6 |
| TAP41 | | 2.46 | 1.83 | 0.74 | 25.8 | 18.8 | 7.0 | 10.2 | 72.8 |
| | **Average** | **2.55** | **1.92** | **0.75** | **24.5** | **19.6** | **4.9** | **10.2** | **79.8** |
| | **St. Dev.** | **0.05** | **0.05** | **0.02** | **2.1** | **2.1** | **1.2** | **1.2** | **4.5** |
| TAR19 | Marly limestone (temple) | 2.69 | 2.60 | 0.96 | 3.5 | 3.3 | 0.3 | 1.2 | 92.9 |
| TASC7 | Metasiltstone (outcrop) | 2.70 | 2.19 | 0.81 | 18.8 | 10.8 | 8.0 | 4.9 | 57.3 |

The investigations showed three main sample populations with significantly different physical-mechanical characteristics and corresponding to the main lithologies, i.e., metadolostone, sandstone, "sandstone" *s.l.* (Figures 11–13). Even within each population, some differences between the subgroups could be observed, with some overlapping trends in physical properties. From these populations we can distinguish the samples of marly limestone and siltstone, which have at least partly different behaviours.

The metasiltstone, represented by a single sample, was not present in the studied monument but its petrophysical characterization could provide a reason why Romans decided not to use this material, outcropping very close to the monument, in building the temple. The sample population of the three types of metadolostones (laminated, brecciated and massive) differ from the other two populations (sandstone and "sandstone" *s.l.*), as they are less porous and more resistant (Figures 11 and 12). The average helium open porosity varied between 8.4 ± 5.7% in laminated metadolostones and 12.9 ± 3.2% in brecciated metadolostones, while in massive metadolostones the average value was 11.6 ± 2.7% (Table 2). The metasiltstone had a porosity value of 18.8%, which is one of the lowest values among metamorphic rocks. The sandstone showed much greater porosity, with an average value of 33.5 ± 5.0%. "Sandstone" *s.l.* had intermediate values between the two previous populations, with an average of 24.5 ± 2.1%. Marly limestone, due to high compactness, showed a low average value of helium open porosity (3.5%) and a slower absorption compared to the other lithologies. Sandstone and "sandstone" *s.l.*, by virtue of their greater open porosity, showed faster water absorption than metadolomite and metasiltstone.

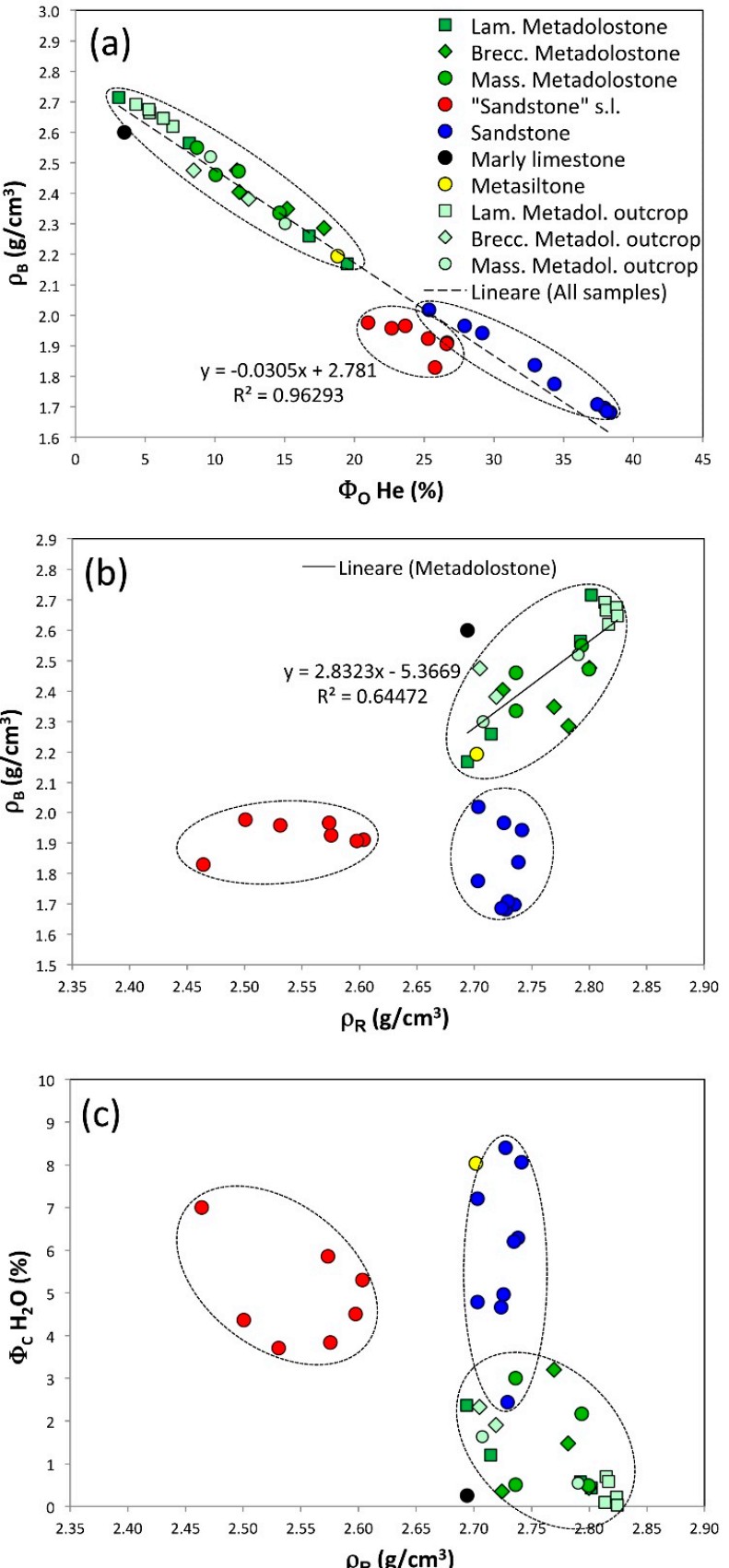

**Figure 11.** (**a**) Helium open porosity ($\Phi_O$ $H_2O$) vs. bulk density ($\rho_B$); (**b**) real density ($\rho_R$) vs. bulk density ($\rho_B$); (**c**) real density ($\rho_R$) vs. water closed porosity ($\Phi_C$ $H_2O$).

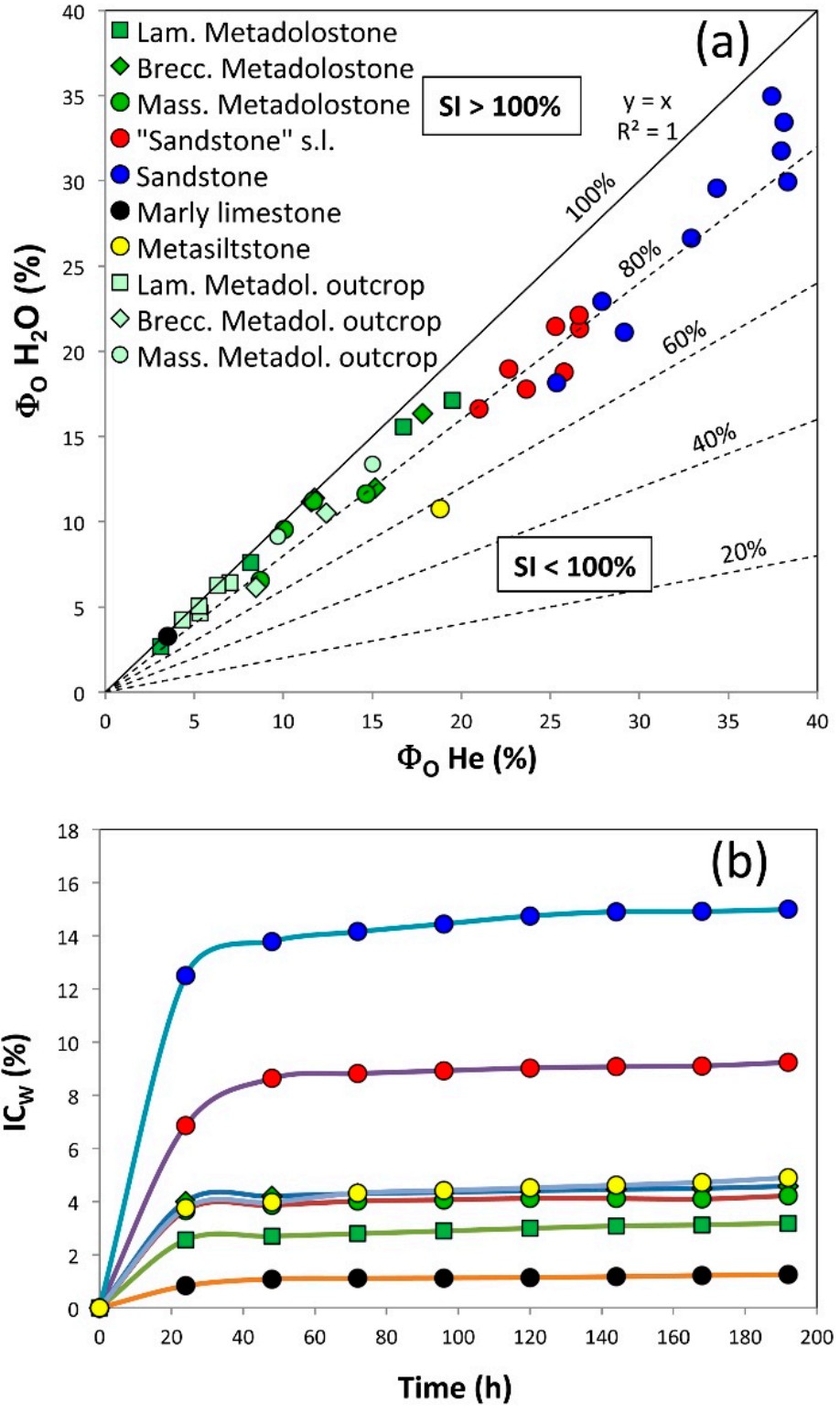

**Figure 12.** (**a**) Helium open porosity ($\Phi_O$ He) vs. water open porosity ($\Phi_O$ $H_2O$); (**b**) kinetic of water absorption shown by imbibition coefficient ($IC_W$) over 192 h.

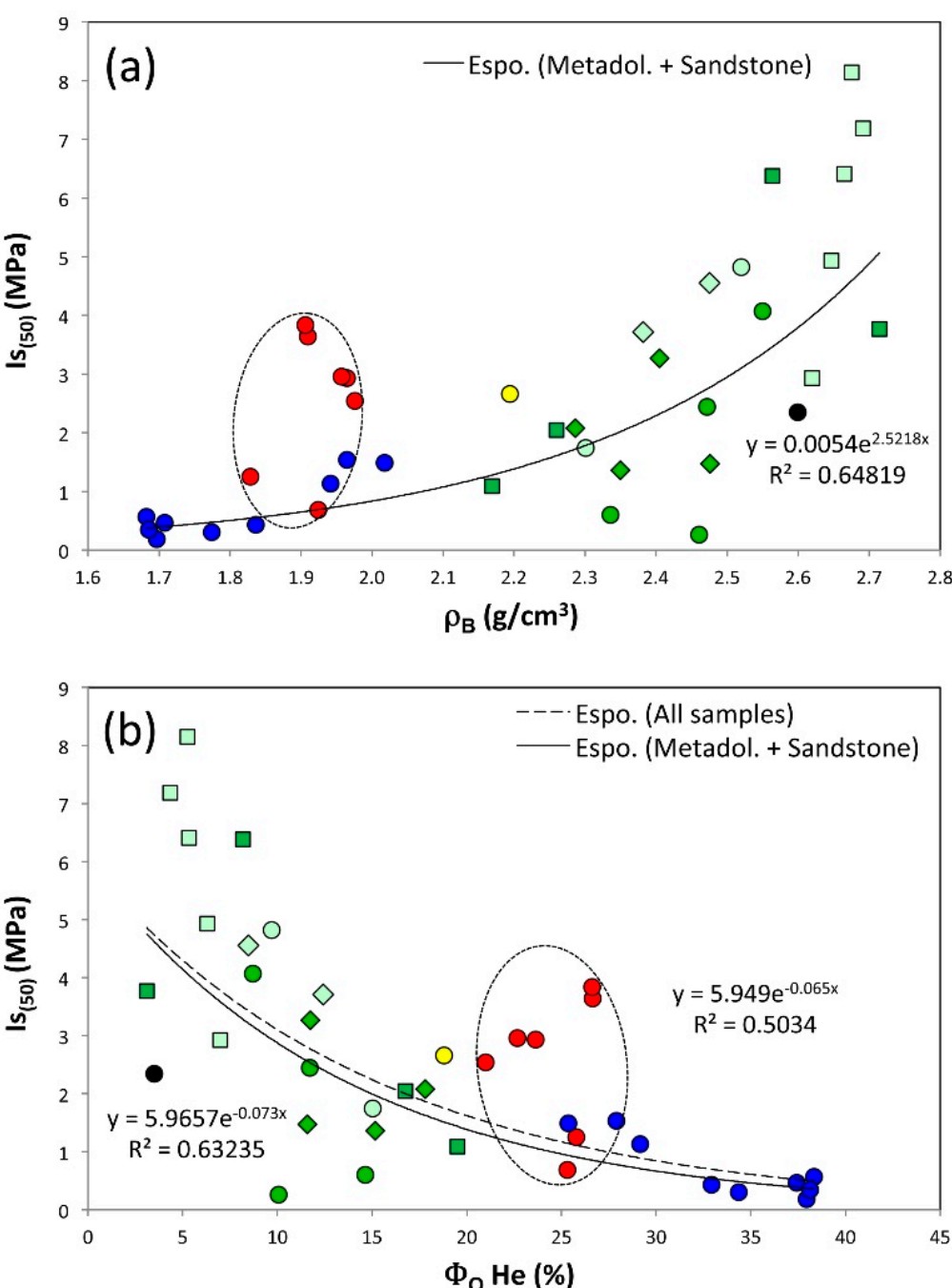

**Figure 13.** (**a**) Bulk density (($\rho_B$) vs. PLT index (Is$_{(50)}$); (**b**) helium open porosity ($\Phi$ He) vs. PLT index (Is$_{(50)}$).

Observing the data as a whole, the bulk density showed a clear linear negative correlation with porosity (Figure 11a) and no evident positive correlation with the real density (Figure 11b) due to the presence of populations of the sandstone and especially "sandstone" *s.l.* samples, which had different physical properties with respect to the metadolostone, showing a clear linear positive correlation coefficient ($R^2 = 0.6$, Figure 11b). The bulk density had average values ranging from a minimum of $2.40 \pm 0.07$ g/cm$^3$ in the samples of brecciated dolomite, an intermediate value of $2.44 \pm 0.10$ g/cm$^3$ in the samples of massive dolomite, to a maximum of $2.56 \pm 0.20$ g/cm$^3$ in the laminated dolomite samples (Table 2). The sandstone showed a much lower value of $1.81 \pm 0.13$ g/cm$^3$, as well as the "sandstone" *s.l.* samples, in which there was an average value of $1.92 \pm 0.05$ g/cm$^3$

(Table 2). The marly limestone, given their low porosity, showed a high bulk density value of 2.60 g/cm$^3$. Metasiltstone had bulk and real densities of 2.19 and 2.70 g/cm$^3$ (Table 2), respectively, and thus represented one of the lowest values among the metamorphic rocks.

The real density, which is mainly influenced by the mineralogical composition, and only to a limited extent by the possible closed porosity (not determined here), showed distinct values between metadolostone and sandstone. In the former, due to the strong presence of the dolomite phase (with a density of 2.84 g/cm$^3$) and secondly of calcite (2.71 g/cm$^3$), this property varied from 2.75 ± 0.04 g/cm$^3$ in brecciated metadolostone, to 2.76 ± 0.04 g/cm$^3$ in massive metadolostone, to 2.79 ± 0.05 g/cm$^3$ in laminated metadolostone.

While sandstone, due to the prevalent presence of quartz (2.65 g/cm$^3$), K-feldspar (2.54–2.56 g/cm$^3$), abundant fossils (2.65–2.71 g/cm$^3$), and only subordinate of phases with high density (e.g., calcite deriving from the "cement" of the matrix), showed a lower average value, equal to 2.73 ± 0.01 g/cm$^3$ (Table 2). The "sandstone" *s.l.* samples were characterized by the lowest real densities, ranging between 2.46 and 2.60 g/cm$^3$, with an average value of 2.55 ± 0.05 g/cm$^3$ (Table 2). This is likely due to the lime-based artificial binder which, during the carbonation process of Ca(OH)$_2$ to form CaCO$_3$ (by CO$_2$ capture from the atmosphere), developed a significant amount of intraphase closed microporosity that affected the volume measurement.

The marly limestones, due to a different mineralogy mainly characterized by carbonate phases, organic compounds and low presence of clay, had the lowest average real density compared to all other samples, probably due to the greater presence of microporosity closed to helium (not interconnected).

The compactness coefficient (C), calculated as the ratio between the bulk density and the real density, inversely proportional to the porosity, was quite homogeneous among the three lithotypes of metadolostone as evidenced by the average values of 0.92, 0.87, 0.88, respectively (Table 2), while the sandstone samples, due to a decidedly higher porosity than the metadolostone samples, showed lower compactness index values, with an average value of 0.66. The artificial "sandstone" *s.l.* samples, due to their different composition, had an intermediate average value between the previous values, equal to 0.75 (Table 2). Marlylimestones, due to their very low porosity (<4%), showed the highest compactness index (0.96). Metasiltstone had a C value of 0.81, the lowest among metamorphic rocks.

Among the hydraulic properties, the coefficient of imbibition expressed in weight (IC$_W$) followed the trend shown by the porosity open to water (Figure 12a), with which a clear positive correlation, with values respectively of 3.2 ± 2.5%, 4.7 ± 1.5%, 4.2 ± 1.1% for metadolostones, and 15.5 ± 4.2% for sandstone (Table 2) was found. Artificial "sandstone" *s.l.* had a value of 10.2 ± 1.2%. The marly limestones, having a low open porosity, showed a low value of the imbibition coefficient (1.2%) compared to the other average values observed.

The porosity closed to water, calculated as the difference between the absolute values of open porosity to helium and to water, had quite high values in sandstone and synthetic "sandstone" *s.l.*, respectively of 5.9 ± 1.9%, 4.9 ± 1.2% (Table 2). Due to its intra-crystalline porosity more interconnected with the circulating fluids, metadolostone showed significantly lower averages of 0.7 ± 0.7%, 1.6 ± 1.1%, 1.4 ± 1.1% (Table 2). Similarly, marly limestone had an even lower closed porosity (0.3%).

The saturation index, which expresses the volume of absorbed water compared to the volume of open pores available (i.e., difference between helium open porosity and water open porosity), and which is generally affected by the pore geometry (i.e., size and tortuosity), shows a weakly negative correlation with open porosity (Figure 12a). This parameter averaged between 87 and 92% in the less porous samples (metadolostones and marly limestones), while it was around 80–82% in the sandstone samples (Table 2). Metasiltstone significantly differed from the other samples, showing a saturation index lower than 60%. The behaviour of the saturation index of lithologies can also be seen graphically in Figure 12a, which shows the porosity open to helium and the porosity open

to water, in which it is observed that the population of the metadolostones samples (green in colour), are close to the line at 100%. In Figure 12b the kinetic of water absorption for total immersion is shown (Table 3).

**Table 3.** Data of kinetic water absorption during the time for 192 h.

| Sample | Lithology | Water Immersion Absorption (g) | | | | | | | |
|---|---|---|---|---|---|---|---|---|---|
| | | 24 h | 48 h | 72 h | 96 h | 120 h | 144 h | 168 h | 192 h |
| TAP33 | Laminated metadolostone (from temples) | 0.89 | 0.92 | 0.94 | 0.96 | 0.97 | 1.11 | 0.97 | 0.98 |
| TAR3 | | 6.23 | 6.36 | 6.60 | 6.65 | 6.69 | 6.72 | 6.73 | 6.86 |
| TAR5 | | 4.85 | 5.38 | 5.39 | 6.04 | 6.70 | 7.37 | 7.84 | 7.85 |
| TAC2 | Lam. dol. (quarry) | 2.64 | 2.75 | 3.01 | 3.00 | 2.99 | 2.95 | 2.94 | 2.96 |
| TASC3 | Laminated metadolostone (from outcrops) | 1.35 | 1.35 | 1.51 | 1.51 | 1.51 | 1.49 | 1.53 | 1.57 |
| TASC4 | | 1.44 | 1.49 | 1.60 | 1.63 | 1.66 | 1.69 | 1.70 | 1.73 |
| TASC5 | | 2.02 | 2.14 | 2.19 | 2.24 | 2.30 | 2.29 | 2.26 | 2.44 |
| TASC6 | | 1.56 | 1.65 | 1.67 | 1.74 | 1.81 | 1.80 | 1.79 | 1.88 |
| TASC8 | | 1.97 | 2.17 | 2.19 | 2.23 | 2.28 | 2.28 | 2.30 | 2.36 |
| | **Average** | **2.55** | **2.69** | **2.79** | **2.89** | **2.99** | **3.08** | **3.12** | **3.18** |
| TAP45 | Brecciated metadolostone (from temples) | 4.42 | 4.44 | 4.47 | 4.58 | 4.69 | 4.67 | 4.62 | 4.73 |
| TAR2 | | 4.36 | 4.40 | 4.42 | 4.59 | 4.76 | 4.89 | 5.05 | 5.08 |
| TAR4 | | 3.68 | 3.95 | 4.10 | 4.19 | 4.27 | 4.33 | 4.37 | 4.48 |
| TAR7 | | 6.22 | 6.57 | 6.77 | 6.83 | 6.89 | 6.97 | 6.99 | 7.12 |
| TAC1 | Brecciated metadolostone (from quarry) | 1.11 | 1.55 | 1.69 | 1.58 | 1.46 | 1.49 | 1.55 | 1.62 |
| TAC3 | | 4.22 | 4.26 | 4.28 | 4.30 | 4.32 | 4.35 | 4.37 | 4.40 |
| | **Average** | **4.00** | **4.20** | **4.29** | **4.34** | **4.40** | **4.45** | **4.49** | **4.57** |
| TAR1 | Massive metadolostone (from Roman temple) | 4.21 | 4.48 | 4.68 | 4.85 | 5.01 | 4.92 | 4.76 | 4.91 |
| TAR20 | | 3.73 | 3.77 | 4.14 | 4.16 | 4.19 | 4.17 | 4.17 | 4.53 |
| TAR22 | | 2.20 | 2.41 | 2.42 | 2.48 | 2.55 | 2.55 | 2.56 | 2.56 |
| TAR49 | | 3.46 | 3.57 | 3.69 | 3.72 | 3.74 | 3.77 | 3.79 | 3.87 |
| TASC1 | Massive metadolostone (from outcrops) | 5.33 | 5.52 | 5.72 | 5.74 | 5.76 | 5.77 | 5.75 | 5.79 |
| TASC2 | | 3.04 | 3.38 | 3.40 | 3.42 | 3.44 | 3.48 | 3.52 | 3.62 |
| | **Average** | **3.66** | **3.85** | **4.01** | **4.06** | **4.11** | **4.11** | **4.09** | **4.21** |
| TAP23 | Sandstone (from temples) | 12.78 | 13.35 | 13.40 | 14.04 | 14.67 | 15.00 | 13.98 | 14.21 |
| TAP27 | | 7.76 | 9.20 | 9.24 | 10.04 | 10.84 | 11.13 | 11.43 | 11.61 |
| TAP31 | | 15.93 | 17.00 | 17.00 | 17.22 | 17.44 | 17.50 | 17.54 | 17.58 |
| TAP35 | | 6.25 | 7.36 | 7.92 | 8.01 | 8.10 | 8.32 | 8.61 | 8.93 |
| TAP36 | | 9.72 | 9.98 | 10.04 | 10.10 | 10.17 | 10.25 | 10.29 | 10.68 |
| TAP42 | | 15.61 | 16.53 | 16.75 | 16.96 | 17.17 | 17.29 | 17.47 | 17.48 |
| TAP48 | | 13.36 | 14.64 | 15.25 | 15.59 | 15.93 | 16.08 | 16.21 | 16.24 |
| TAR14 | | 17.28 | 19.31 | 19.65 | 19.85 | 20.05 | 20.11 | 20.15 | 20.18 |
| TAR50 | | 13.76 | 16.68 | 18.09 | 18.18 | 18.28 | 18.42 | 18.51 | 17.98 |
| | **Average** | **12.49** | **13.78** | **14.15** | **14.44** | **14.74** | **14.90** | **14.91** | **14.99** |
| TAP30 | "Sandstone" s.l. (from Punic temple) | 9.39 | 10.45 | 10.53 | 10.64 | 10.75 | 10.83 | 10.90 | 11.13 |
| TAP32 | | 4.22 | 4.42 | 4.48 | 4.63 | 4.78 | 4.76 | 4.75 | 4.95 |
| TAP37 | | 7.49 | 7.79 | 8.00 | 8.06 | 8.13 | 8.15 | 8.15 | 8.37 |
| TAP38 | | 1.90 | 11.39 | 11.50 | 11.52 | 11.54 | 11.56 | 11.56 | 11.57 |
| TAP39 | | 7.43 | 8.12 | 8.45 | 8.61 | 8.77 | 8.90 | 8.96 | 9.02 |
| TAP40 | | 8.80 | 9.09 | 9.30 | 9.39 | 9.47 | 9.50 | 9.54 | 9.64 |
| TAP41 | | 8.76 | 9.12 | 9.44 | 9.58 | 9.73 | 9.78 | 9.84 | 9.92 |
| | Average | 6.86 | 8.63 | 8.81 | 8.92 | 9.02 | 9.07 | 9.10 | 9.23 |
| TAR19 | Marly limestone | 0.82 | 1.08 | 1.11 | 1.12 | 1.14 | 1.17 | 1.21 | 1.25 |
| TASC7 | Metasiltstone | 3.74 | 3.97 | 4.32 | 4.41 | 4.51 | 4.61 | 4.72 | 4.89 |

From the physical-mechanical point of view, considering the PLT resistance data of metadolostone and sandstone samples (Table 4), a general positive exponential correlation was observed between the $Is_{(50)}$ index (MPa) versus bulk density (Figure 13a) and the $Is_{(50)}$ index versus open porosity to helium (Figure 13b), with determination coefficients $R^2$ equal to 0.65 and 0.63, respectively. Considering all samples (including the "sandstone" *s.l.*) the $R^2$ value decreased due to the different trend shown by the population. Therefore, the highest values of the PLT index were found in the metadolostone samples (4.76 ± 2.45, 2.74 ± 1.30, 2.32 ± 1.84 MPa, Table 4), which had lower porosity, while the lowest values, with an average of 0.72 ± 0.52 MPa, occurred in the most porous samples (sandstone), as they had a carbonate "cement" of poor compactness and resistance. The "sandstone" *s.l.* samples, on the other hand, by virtue of a carbonation of lime-based binder, showed a much higher resistance value than that of sandstone, with an average value of 2.55 ± 1.18 MPa (Table 4). The values of the standard deviation show a greater variability of the mechanical resistance in metadolostone samples due to their more brittle behaviour compared to sandstone and "sandstone" *s.l.*, which, on the other hand, showed better absorption and distribution of mechanical stresses.

**Table 4.** Point Load Test data. Symbol legend: W = width of specimen; H = height of specimen; P = breaking load; $D_e$ = equivalent diameter; Is = strength index; F = correction factor; $Is_{(50)}$ = Point Load Test index normalised to cylinder specimen with diameter of 50 mm; $R_C$ = indirectly calculated compression strength; $R_T$ = indirectly calculated tensile strength.

| Sample | Lithology | W | H | P | $D_e{}^2$ | Is | F | $Is_{(50)}$ | $R_C$ | | $R_T$ | |
|---|---|---|---|---|---|---|---|---|---|---|---|---|
| | | mm | mm | kN | mm² | N/mm² | / | MPa | MPa | kg/cm² | MPa | kg/cm² |
| TAP32 | Laminated metadolostone (from temples) | 17.6 | 10.0 | 1.45 | 223.5 | 6.49 | 0.58 | 3.77 | 75.4 | 768.6 | 3.0 | 30.7 |
| TAR3 | | 14.9 | 12.0 | 0.80 | 228.0 | 3.51 | 0.58 | 2.05 | 40.9 | 417.4 | 1.6 | 16.7 |
| TAR5 | | 18.1 | 15.5 | 0.60 | 356.2 | 1.68 | 0.65 | 1.09 | 21.7 | 221.6 | 0.9 | 8.9 |
| TAC2 | Lam. dol. (quarry) | 18.5 | 11.5 | 2.85 | 270.9 | 10.52 | 0.61 | 6.38 | 127.6 | 1301.4 | 5.1 | 52.1 |
| TASC3 | Laminated metadolostone (from outcrops) | 17.7 | 15.5 | 3.90 | 348.3 | 11.20 | 0.64 | 7.19 | 143.7 | 1465.5 | 5.7 | 58.6 |
| TASC4 | | 16.2 | 7.5 | 1.85 | 154.2 | 12.00 | 0.53 | 6.41 | 128.2 | 1307.1 | 5.1 | 52.3 |
| TASC5 | | 17.7 | 8.5 | 1.00 | 191.6 | 5.22 | 0.56 | 2.93 | 58.6 | 597.3 | 2.3 | 23.9 |
| TASC6 | | 17.8 | 15.5 | 4.45 | 351.3 | 12.67 | 0.64 | 8.15 | 162.9 | 1661.3 | 6.5 | 66.5 |
| TASC8 | | 18.9 | 15.0 | 2.75 | 361.0 | 7.62 | 0.65 | 4.93 | 98.6 | 1005.2 | 3.9 | 40.2 |
| | **Average** | **17.5** | **12.3** | **2.18** | **276.1** | **7.88** | **0.60** | **4.76** | **95.3** | **971.7** | **3.8** | **38.9** |
| | **St. Dev.** | **1.2** | **3.2** | **1.38** | **80.3** | **3.94** | **0.04** | **2.45** | **48.9** | **499.1** | **2.0** | **20.0** |
| TAP45 | Brecciated metadolostone (from temples) | 18.7 | 15.5 | 1.85 | 368.1 | 5.03 | 0.65 | 3.27 | 65.3 | 666.1 | 2.6 | 26.6 |
| TAR2 | | 17.4 | 9.5 | 0.50 | 210.5 | 2.38 | 0.57 | 1.36 | 27.2 | 277.6 | 1.1 | 11.1 |
| TAR4 | | 17.5 | 12.0 | 0.65 | 267.4 | 2.43 | 0.60 | 1.47 | 29.4 | 299.8 | 1.2 | 12.0 |
| TAR7 | | 16.8 | 8.0 | 0.65 | 170.9 | 3.80 | 0.55 | 2.08 | 41.6 | 424.2 | 1.7 | 17.0 |
| TAC1 | Brecciated metadolostone (from quarry) | 16.3 | 10.0 | 1.65 | 206.9 | 7.97 | 0.57 | 4.55 | 91.0 | 928.4 | 3.6 | 37.1 |
| TAC3 | | 19.5 | 13.0 | 1.90 | 322.8 | 5.89 | 0.63 | 3.71 | 74.3 | 757.4 | 3.0 | 30.3 |
| | **Average** | **17.7** | **11.3** | **1.20** | **257.7** | **4.58** | **0.60** | **2.74** | **54.8** | **558.9** | **2.2** | **22.4** |
| | **St. Dev.** | **1.2** | **2.7** | **0.66** | **76.0** | **2.17** | **0.04** | **1.30** | **26.0** | **265.3** | **1.0** | **10.6** |
| TAR1 | Massive metadolostone (from Roman temple) | 19.4 | 10.0 | 0.25 | 247.3 | 1.01 | 0.59 | 0.60 | 12.0 | 122.5 | 0.5 | 4.9 |
| TAR20 | | 16.2 | 11.0 | 0.95 | 226.5 | 4.19 | 0.58 | 2.44 | 48.9 | 498.3 | 2.0 | 19.9 |
| TAR22 | | 19.3 | 10.5 | 1.75 | 258.0 | 6.78 | 0.60 | 4.07 | 81.4 | 829.8 | 3.3 | 33.2 |
| TAR49 | | 18.9 | 15.5 | 0.15 | 372.0 | 0.40 | 0.65 | 0.26 | 5.3 | 53.6 | 0.2 | 2.1 |
| TASC1 | Massive metadolostone (from outcrops) | 18.5 | 10.0 | 0.70 | 235.9 | 2.97 | 0.59 | 1.74 | 34.9 | 355.8 | 1.4 | 14.2 |
| TASC2 | | 18.0 | 12.5 | 2.25 | 286.5 | 7.85 | 0.61 | 4.82 | 96.5 | 983.8 | 3.9 | 39.4 |

**Table 4.** *Cont.*

| Sample | Lithology | W | H | P | $D_e^2$ | Is | F | Is$_{(50)}$ | R$_C$ | | R$_T$ | |
|---|---|---|---|---|---|---|---|---|---|---|---|---|
| | | mm | mm | kN | mm² | N/mm² | / | MPa | MPa | kg/cm² | MPa | kg/cm² |
| | **Average** | **18.4** | **11.6** | **1.01** | **271.0** | **3.87** | **0.61** | **2.32** | **46.5** | **474.0** | **1.9** | **19.0** |
| | **St. Dev.** | **1.2** | **2.1** | **0.84** | **53.6** | **3.02** | **0.03** | **1.84** | **36.7** | **374.5** | **1.5** | **15.0** |
| TAP23 | | 20.5 | 7.5 | 0.15 | 195.3 | 0.77 | 0.56 | 0.43 | 8.7 | 88.3 | 0.3 | 3.5 |
| TAP27 | | 20.2 | 15.0 | 0.90 | 384.8 | 2.34 | 0.66 | 1.53 | 30.7 | 313.1 | 1.2 | 12.5 |
| TAP31 | | 20.5 | 13.5 | 0.10 | 351.9 | 0.28 | 0.64 | 0.18 | 3.7 | 37.3 | 0.1 | 1.5 |
| TAP35 | | 16.2 | 11.5 | 0.60 | 237.2 | 2.53 | 0.59 | 1.49 | 29.8 | 303.7 | 1.2 | 12.1 |
| TAP36 | Sandstone (from temples) | 19.7 | 15.0 | 0.65 | 375.3 | 1.73 | 0.65 | 1.13 | 22.6 | 230.5 | 0.9 | 9.2 |
| TAP42 | | 20.9 | 15.5 | 0.35 | 412.5 | 0.85 | 0.67 | 0.57 | 11.3 | 115.4 | 0.5 | 4.6 |
| TAP48 | | 20.3 | 12.0 | 0.15 | 309.8 | 0.48 | 0.63 | 0.30 | 6.1 | 61.7 | 0.2 | 2.5 |
| TAR14 | | 19.2 | 14.0 | 0.25 | 341.8 | 0.73 | 0.64 | 0.47 | 9.3 | 95.3 | 0.4 | 3.8 |
| TAR50 | | 18.5 | 11.0 | 0.15 | 258.4 | 0.58 | 0.60 | 0.35 | 7.0 | 71.0 | 0.3 | 2.8 |
| | **Average** | **19.5** | **12.8** | **0.37** | **318.6** | **1.14** | **0.63** | **0.72** | **14.3** | **146.3** | **0.6** | **5.9** |
| | **St. Dev.** | **1.5** | **2.6** | **0.28** | **73.8** | **0.84** | **0.03** | **0.52** | **10.5** | **106.8** | **0.4** | **4.3** |
| TAP30 | | 17.0 | 8.0 | 1.15 | 173.2 | 6.64 | 0.55 | 3.64 | 72.8 | 742.8 | 2.9 | 29.7 |
| TAP33 | | 18.8 | 11.0 | 0.30 | 263.7 | 1.14 | 0.60 | 0.69 | 13.7 | 139.9 | 0.5 | 5.6 |
| TAP37 | "Sandstone" s.l. (from Punic temple) | 17.3 | 12.5 | 1.15 | 275.3 | 4.18 | 0.61 | 2.54 | 50.8 | 518.5 | 2.0 | 20.7 |
| TAP38 | | 17.4 | 14.0 | 1.90 | 309.3 | 6.14 | 0.62 | 3.84 | 76.8 | 782.9 | 3.1 | 31.3 |
| TAP39 | | 15.0 | 12.0 | 1.15 | 229.2 | 5.02 | 0.58 | 2.93 | 58.6 | 597.8 | 2.3 | 23.9 |
| TAP40 | | 21.8 | 12.5 | 1.60 | 347.0 | 4.61 | 0.64 | 2.96 | 59.1 | 603.1 | 2.4 | 24.1 |
| TAP41 | | 19.5 | 12.0 | 0.60 | 297.6 | 2.02 | 0.62 | 1.25 | 25.0 | 254.7 | 1.0 | 10.2 |
| | **Average** | **18.1** | **11.7** | **1.12** | **270.7** | **4.25** | **0.60** | **2.55** | **51.0** | **520.0** | **2.0** | **20.8** |
| | **St. Dev.** | **2.2** | **1.9** | **0.55** | **56.8** | **2.03** | **0.03** | **1.18** | **23.6** | **240.3** | **0.9** | **9.6** |
| TAR19 | Marly limestone | 17.0 | 11.0 | 0.95 | 238.5 | 3.98 | 0.59 | 2.35 | 47.0 | 478.9 | 1.9 | 19.2 |
| TASC7 | Metasiltstone | 19.1 | 15.0 | 1.50 | 365.3 | 4.11 | 0.65 | 2.66 | 53.3 | 543.3 | 2.1 | 21.7 |

## 6. Discussion

The results of the mineralogical and petrographic analysis show that in the construction of the temples two main lithologies of different nature with different compositional characteristics and physical-mechanical behaviour were used: a sandstone and a metadolostone represented by three different subfacies characterized by different microstructures and textures (i.e., laminated, brecciated and massive). Based on the casual distribution of the diverse metadolostone facies ashlars in the different parts of the temple, as a ground hypothesis it seems that those blocks were used randomly. Nonetheless the multiple quarry front dug contemporaneously by the Romans seems to suggest a specific need for every facies.

Both laminated and brecciated metadolostones were used for the construction of the Punic temple. In more detail, the laminated metadolostone (TAP32 sample, Figure 5) was compositionally similar to the TAC2 sample belonging to an extraction front of the ancient quarry (called "Canali Bingias", Figure 14) located north/northeast of the Antas temple (at a distance of about 600 m as the crow flies) and also to the metadolostone samples (TASC3, TASC4, TASC5, TASC6, TASC8, Figure 14) outcropping at the northeast and southwest of the temple. The brecciated metadolostone (TAP45 sample, Figure 5) instead showed a clear petrographic resemblance to the samples of other extraction fronts of the ancient quarry (TAC1, TAC3 samples, Figure 14).

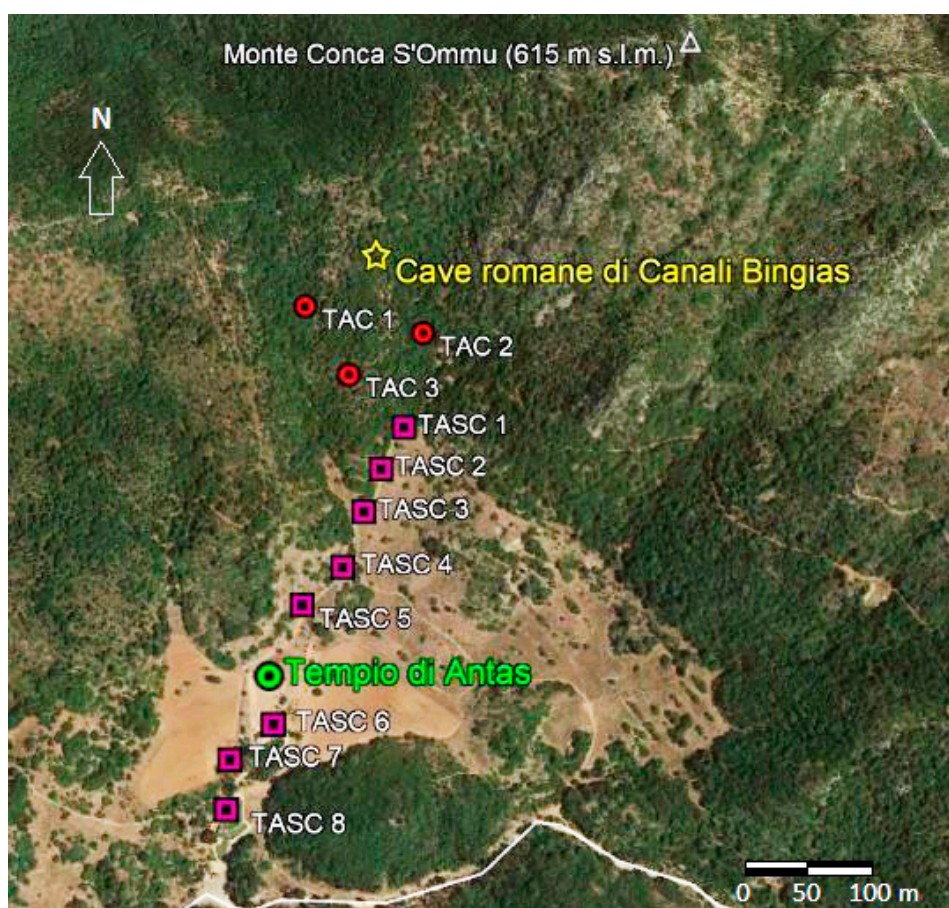

**Figure 14.** Aerial view of the area around the archaeological site of Antas (from Google Earth, modified, 2015). The yellow star indicates the position of the Roman quarries in locality called "Canali Bingias; the red circles indicate the sampling of the lithologies (samples TAC1, TAC2, TAC3) at south of ancient quarries; the purple squares indicate the sampling of outcrops (from TASC1 to TASC5) along the path that leads from the ancient quarries up to the temple, and the outcrops at South of Antas site (samples TASC6, TASC7, TASC8).

As regards the Roman temple building, the laminated metadolostones (TAR3, TAR5 samples, Figure 6) and the brecciated metadolostones (TAR2, TAR4, TAR7, TAR45 samples, Figure 6) show a similarity to the extraction fronts already used in the Punic phase described above, while the massive metadolostones (TAR1, TAR20, TAR22, TAR49 samples, Figure 6) have characteristics similar to the TASC1, TASC2 samples belonging to an outcrop that was opened and exploited in the Roman phase, located at South of the ancient quarry. In this latter, including the TAC1, TAC2, TAC3 samples (Figure 14), where both laminated and brecciated metadolostones crop out, today traces of processing (e.g., furrows, water drains) and systems can be still observed in their entirety of Roman technology extraction of ashlars with various sizes. This archaeological evidence, associated with the results of the petrographic investigations, confirm that the ancient Roman quarries were actually the main supply point of the geomaterials in the Roman phase for the construction of the second temple building. However, other outcrops (signed with TASC, Figure 14) closer to the two temples, were also exploited both in the Punic and Roman phases. The remains of these extraction fronts have not been found up to now, probably because they have been obliterated by vegetation and/or because of their small dimensions. Therefore, it is plausible that the supply of the construction material rested on an extraction area with a greater extension than that currently observed in the area of the Roman quarry.

On the podium of the Roman temple, in the limit that connects the pronaos to the cell, a small "threshold" consisting of a marly limestone was identified but was not found

anywhere else in the templar building. Its different origin compared to the other lithologies has not been defined at the moment. It could have had a dividing function between two different environments in the same way in which slates were used for the separation of the environments, or it could be a substitute material deriving from a subsequent restoration (historical ? or recent ? to be investigated), for which it is not possible today to define its exact function. It is worthy of note that its very low porosity makes this rock suitable for the threshold function, exposed to weathering but also to treading. This consideration suggests a precise material selection by Roman builders rather than a subsequent restoration that was unlikely performed with a totally different material. As regards the Punic temple, it was found that the materials used for its construction mainly consist of fossiliferous sandstone characterized by a coarser and incoherent sediment, less mature than the other lithology, with generally subrounded clasts and a predominantly calcitic cement. This lithology does not crop out in the immediate vicinity of the archaeological site but can be found in the central-western coast of the island, which was largely exploited in the Punic and Roman phases.

Historical mining activity shows a wide use of this sandstone which, being largely outcropping on the coast, was easy to extract and transport by sea. Having sufficient resistance to weathering and great workability, this lithology together with its ancient quarries, is commonly found from the south of Sardinia (e.g., at the archaeological area of Nora where at "Fradis Minoris" [20]), to southwest (e.g., Piscinnì Roman quarries, near Teulada), where ancient extractive fronts in sandstone outcrops (e.g., the formation called "Panchina Tirreniana" AA [20]) are found. Going northward, there are other processing fronts in sandstone outcrops from the Punic and Roman times in the coast immediately west of the Antas area and further North (e.g., s'Enna 'e s'Arca quarry, Figures 4 and 15).

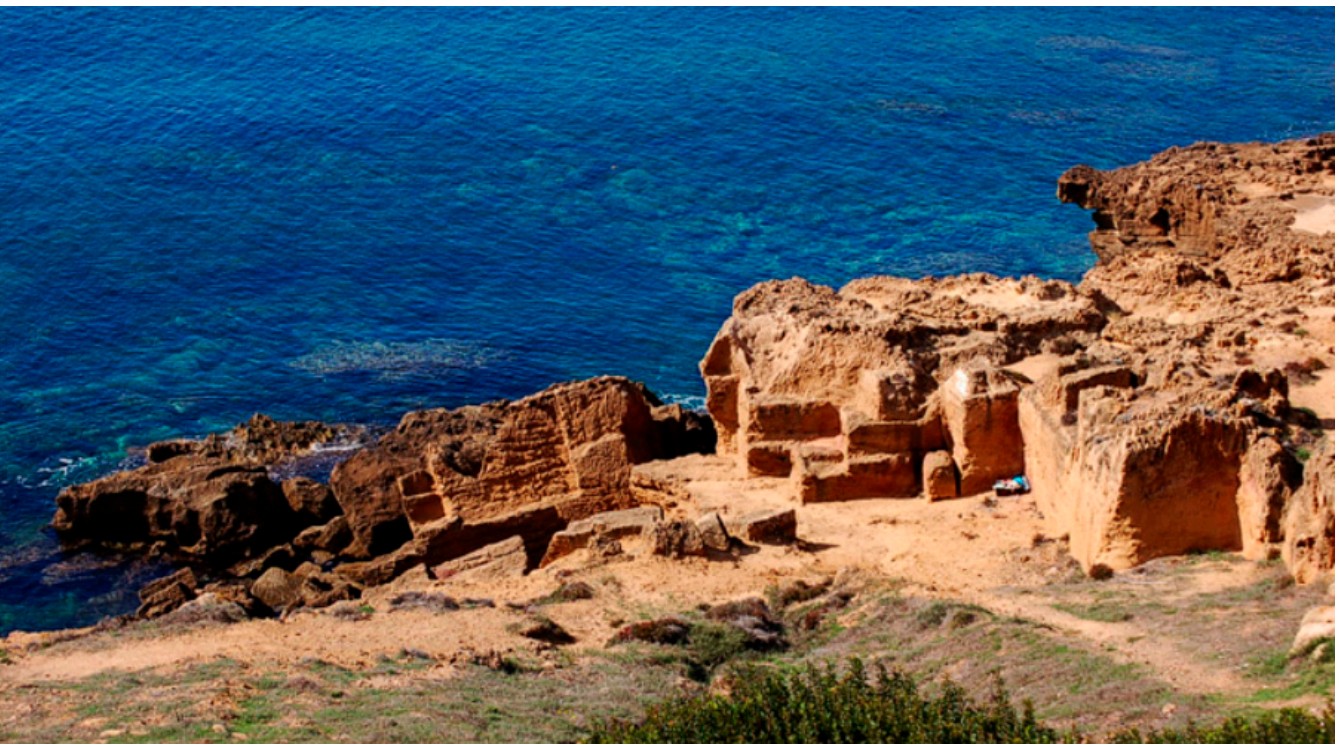

**Figure 15.** Ancient quarry of sandstone (called "s'Enna 'e s'Arca") used in Punic and Roman times located on the western coast of Sardinia (see Figure 4).

Moreover, another peculiar lithoid material was found as a building stone in the Punic temple. Preliminarily this material was identified as a generic sandstone but detailed mineralogical-petrographic analysis showed that it was not a natural stone and it has

been called sandstone-like rock (i.e., "sandstone" *sensu lato*). In fact, under the microscope it shows characteristics of high compositional immaturity (heterogeneous association of quartz, feldspar, lithics of various origins) coupled with contrasting textural maturity aspects: grains of all types with matching grain size, but at the same time showing similar rounding and sphericity qualitative indexes, floating in a cemented, hardened carbonate matrix. This is incompatible even with typical textural inversion phenomena [80]. These sedimentary features are not found in nature together and are irreconcilable with natural flow conditions and routing sediment systems [89], or with post-depositional reworking processes. Therefore, this is definitely an artificial geomaterial probably made during the massive restoration interventions of the two temples that took place in the 20th century by the Superintendence of Archaeological Heritage of Cagliari and the Province, as was also done for the reconstruction of some degraded ashlars of metadolostones used for the columns, bases and capitals of the Roman temple.

The results of the physical-mechanical analyses on geomaterials highlight the different behaviour of the analysed materials (metadolostone, sandstone, metasiltstone, marly limestone, "sandstone" *s.l.*), substantially linked to the petrogenetic aspects of these rocks (and artificial materials), and subject to the degradation processes that occurred following their installation in the original construction phase. Due to their different textural characteristics, laminated, brecciated and massive metadolostones are less porous (on average 5–13%) than natural sandstones (25–39%) and man-made "sandstone" *s.l.* (21–27%). In fact, metadolostone rocks have a mainly intracrystalline porosity (between dolomite and calcite crystals), substantially primary and with a presumably smaller diameter (<100 microns), while sandstone has an intragranular porosity (mixed intraclast and intracrystal), distributed in the carbonate cement matrix. This porosity is both primary and secondary induced by dissolution processes due to chemical-physical decay. On the surface of stone there is occasionally a filling of porosity by secondary phases of reprecipitation with a carbonate composition.

The lower porosity of metadolostone rocks certainly results in a higher average mechanical strength (e.g., in laminated ones with an average compressive strength of $972 \pm 499\,\mathrm{kg/cm^2}$) than sandstone ($146 \pm 107\,\mathrm{kg/cm^2}$). However, different from the latter, there was a very high variability of the data due to fragile physical-mechanical behaviour of the metadolostone, which involves variable micro-fracturing (sometimes even at the macro-scale according to the anisotropy planes) evident in the material when subjected to stress.

These petrophysical aspects meant that in the Roman construction phase at the Antas site, characterized by more advanced mining and processing technology of the stone, better rock materials were used, but those more difficult to work such as the metadolostones, while in the Punic phase there was prevalent use of sandstone which, as is well known, has easier workability and sufficient mechanical strength to be used as a construction material. It is interesting to note that neither Romans nor Punic builders considered using the metasiltstone outcropping close to the Temples. This could be due to a precise architectural-aesthetic choice but could also be explained by petrophysical properties. Indeed, physical and mechanical tests showed that this material is stronger and less porous than the sandstone used during Punic phase but weaker compared to metadolostone used by Romans. Thus, if it was not a stylistic choice (we cannot exclude this hypothesis), it can be argued that this rock was too strong for Punic stoneworkers (as well as metadolostone) but too weak for Roman workers.

A last consideration about the other topic of this paper, i.e., the conservation state of the building stones, can be partially addressed. The petrophysical data of dolostone showed that the samples belonging to the Roman temple commonly have a greater porosity than those from the quarry's outcrops. The PLT results also highlighted a difference between samples of dolostone from natural outcrops, mainly having $Is_{(50)} > 3$ MPa, and samples from the Roman temple, mainly below this value. These data clearly indicate that building stones were more affected by weathering processes (rainfall producing the

chemical dissolution of carbonate matrix, thermal stress, etc.) than the outcrop rocks protected by vegetation covers.

Regarding the sandstone used in the Punic temple, it is not possible to assess the decay state without having a term of comparison to the quarry from which the ashlars were carved. The research of the exact source of this material is now in progress and, if it is positively accomplished, it will be possible to determine the decay state of Punic monument by comparing the petrophysical data presented here with data measured on quarry samples.

## 7. Conclusions

The research allowed us to define the compositional and physical-mechanical characteristics of the metadolostone and sandstone used for the construction the Punic and Roman temples.

The results of investigations highlight that the Punic and the Romans used different lithologies for the construction of the two temples. Although the temple was built on an outcrop of dolomitic rocks, the Punic people, considering the limited size of the temple, mainly used sandstone rocks, which are more easily workable and rough-hewn into ashlars. These rocks are absent in the immediate vicinity of the Antas Temple but are found abundantly on the west coast of the island, where there are extractive fronts still recognizable today. In fact, the sandstone, together with the arenaceous conglomerate and other calcarenite-type rocks, has been extensively quarried on the coasts in different historical periods and Sardinian Punic-Roman sites (e.g., *Nora*, *Tharros*, southwestern and central-western Sardinia, respectively), due to their easy workability, extractability and transportability by sea. Presumably the sandstone was extracted along the coast of the western coast about 7 km away at the closest point corresponding to the current village of Buggerru (near to the probable localization of *Neapolis* Roman village, Figure 4) with an important Punic-Roman port that supported the important mining activity of *Metalla* in Roman times, was used for the ancient trade of mineral products, and connected with other ports and villages of the Punic-Roman settlements (e.g., *Tharros* in Central-West of Sardinia; *Sulki* in southeast; *Bithia* and *Nora* in the south, Figure 4).

The Romans built a more articulated temple building from an architectural point of view, with valuable technical-stylistic aspects. Given the greater dimensions in plan-volumetric terms and structural complexity of this temple, the Romans looked for a stone that could satisfy the most stringent requirements of physical-mechanical resistance and that was easily available in the surrounding area. By virtue of more refined mining and processing technology than the Punics, they chose a mechanically very competent dolomitic stone emerging a few kilometres from the temple.

The presence of several small quarries suggests the Romans exploited all the quarries simultaneously and had diverse building uses for the different brecciated, laminated, massive metadolostone facies from any quarry. Nonetheless, the absence of real knowledge of the initial status and structure of the temple, and the corresponding use of the different facies, hampers us from understanding what these purposes were.

The laminated facies, which emerges more copiously and near the temple, is the most used, also by virtue of a lower porosity that entails greater mechanical strength. The brecciated facies, the least resistant of the three, is the least used in the construction of the temple. Despite the excellent durability of the metadolostone rock, the one used in the building shows signs of decay, as evidenced by an increase in the porosity open to the water and a consequent lower physical-mechanical resistance. In fact, being a carbonate-based rock ($CaCO_3$, $MgCO_3$) it suffers from the processes of chemical dissolution by weathering.

The study also made it possible to highlight the use of other artificial geomaterials, used during the important restoration interventions that took place in recent decades during which, in addition to significant and articulated work of anastylosis, various decorative and structural elements (e.g., ashlars and architraves) were replaced. The presence of a matrix-indurated cement-based conglomerate, already known and studied by local authorities,

scholars and archaeologists, and made up using fine sand and centimetre fragments of metadolostone rock as aggregate, was recognized. In addition, this study documented the presence (hitherto unknown) of an artificial sandy-conglomeratic stone that is aesthetically very similar to a sandstone rock and that consists of selected quartz + feldspars + lithic fragments sand aggregates with a lime-based binder.

**Author Contributions:** Conceptualization, S.C.; data curation, S.C.; formal analysis, E.G. and D.F.; investigation, E.G., L.G.C. and D.F.; validation, L.G.C.; writing—original draft, S.C.; writing—review and editing, L.G.C. and D.F. All authors have read and agreed to the published version of the manuscript.

**Funding:** This research was funded by PON-AIM Project—"Attraction and International Mobility" (Rif. AIM1890410-3, D.D. n. 407 del 27.02.2018) of Chemical and Geological Sciences Department of Cagliari University—MIUR: Dipartimento per la formazione superiore e per la ricerca Direzione Generale per il coordinamento, la promozione e la valorizzazione della Ricerca.

**Acknowledgments:** We thank the "Soprintendenza Archeologia, Belle Arti e Paesaggio per la città Metropolitana di Cagliari e le province di Oristano e Sud Sardegna", Massimo Casagrande and the people in charge for obtaining the authorization for the materials sampling and the in situ study activities, the "Start Uno" tourist-cultural cooperative of Fluminimaggiore, and the archaeologists: Michela Migaleddu, Alessandra Gaviano and Valentina Tiddia for supporting the collection of historical-archaeological data and the research carried out in the archaeological area.

**Conflicts of Interest:** The authors declare no conflict of interest.

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
