# Peer review of "Mineralogical-Petrographic and Physical-Mechanical Features of the Construction Stones in Punic and Roman Temples of Antas (SW Sardinia, Italy): Provenance of the Raw Materials and Conservation State"

_minerals, doi:10.3390/min11090964_

Round 1

Reviewer 1 Report

The authors present a study of the geomaterials used in the Antas Punic-roman Temples from Sardinia, Italy. The paper is well organised and sound. The information  would clearly be of interest of international public. Before publication I would suggest to consider some issues which are given bellow.

General comments

I am not find sandstone-like rock appropriate term (if artificial - sandstone like material?). Also currently in the paper is stated that the binder is carbonate and that is not possible to state if natural or artificial, in some places authors mention lime-based materials, which is mortars. If lime mortars, this could be recognised. usually lime lumps are present. In my opinion sandstones could be distinguished from artificial materials. And if lime mortars, please consider the physcial-mechanical properties obtained to help you with confirmation if material natural or artificial.

Why for comparison only metadolostone is samples from outcrops? What about sandstones?

Results are presented in tables and figures. In my opinion figures would be sufficient.

Title: please consider to use geomaterials instead of construction stones?

Please, unify terminology: minero-petrographic or mineral-petrographic throughout the manuscript; kinetic of water absorption, imbibition

Petrophysical properties: comparison between samples from temples and outcrops would be suggested

Results on metasiltstone are not discussed in petrophysical properties section. Please, check if data are described for all types of stones

Please, summarize in the conclusion which tape of materials was determined

Minor issues

-line 12: please, add Antas site (SW Sardinia, Italy)

-line 40: please, add datation for Punic phase

-lines44-47: please add references

-line 64: please, add reference

-line 90-conservation-restoration

-line 289-ancinet quarries as well?

-line 335: please, list which physical tests

-line 378; please provide which samples goes for laminated, massive and brecciated dolomites in parenthesis

-lines339-409: please provide % of grains, Only one group of sandstone samples was recognised?

-line 421: carbonate cement or lime binder?

-lines 425-436: please, provide info on the binder, grain sizes, shape, % of grains

-line 445: origin for metasiltstone could be discussed?

-lines 466-468: should be deleted?

-lines 673-679: is then suggested that aggregate was mixed? usually natural aggregate is used from river, ect..,

-lines722-725: please note that not the same elements of the building were sampled from the two temples in order to make a comparison? Could the difference be related to application purposes?

Round 2

Reviewer 2 Report

Please replace in the new line 206 "is" with "are". The Antas Temples are...

No other comments. Very interesting work!